# RoboCerebra: A Large-scale Benchmark for Long-horizon Robotic Manipulation Evaluation

**Songhao Han**[1*]    **Boxiang Qiu**[1*]    **Yue Liao**[2*†]    **Siyuan Huang**[3]
**Chen Gao**[1]    **Shuicheng Yan**[2‡]    **Si Liu**[1‡]

[1]Beihang University    [2]National University of Singapore    [3]Shanghai Jiao Tong University

`robocerebra.github.io`

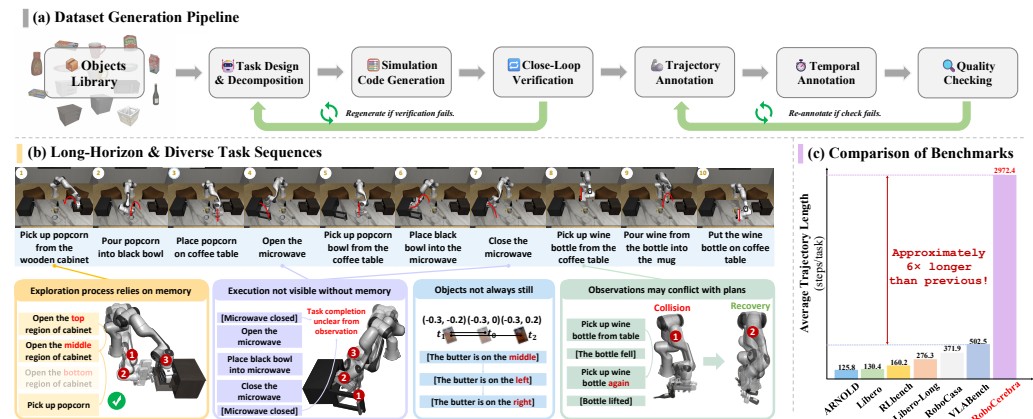

Figure 1: We shift the focus of robotic imitation learning from fast, reactive System 1 behavior to slow, deliberative System 2 reasoning. To support this, we introduce *RoboCerebra*, a benchmark centered on long-horizon tasks composed of extended subtask sequences. (a) A top-down data generation pipeline uses an LLM to produce high-level task instructions and decompose them into subtasks. Human operators execute these in simulation to collect trajectories, with multi-stage verification ensuring quality and semantic consistency. (b) A dataset example showing a long, fine-grained subtask sequence under dynamically changing visual conditions. (c) RoboCerebra features significantly longer trajectories, approximately *6×* those in existing robotic manipulation benchmarks.

## Abstract

Recent advances in vision-language models (VLMs) have enabled instruction-conditioned robotic systems with improved generalization. However, most existing work focuses on reactive System 1 policies, underutilizing VLMs' strengths in semantic reasoning and long-horizon planning. These System 2 capabilities—characterized by deliberative, goal-directed thinking—remain underexplored due to the limited temporal scale and structural complexity of current benchmarks. To address this gap, we introduce RoboCerebra, a benchmark for evaluating high-level reasoning in long-horizon robotic manipulation. RoboCerebra includes: (1) a large-scale simulation dataset with extended task horizons and diverse subtask sequences in household environments; (2) a hierarchical framework combining a high-level VLM planner with a low-level vision-language-action (VLA) controller; and (3) an evaluation protocol targeting planning, reflection, and memory through

---

[*]Equal contribution

[†]Project Leader

[‡]Corresponding Author

39th Conference on Neural Information Processing Systems (NeurIPS 2025) Track on Datasets and Benchmarks.

Table 1: **Comparison of Benchmarks.** Our *RoboCerebra* benchmark is designed to evaluate System 2 capabilities in robotic manipulation. It is designed to generate long-horizon tasks with large language models (LLMs), enriched with human-collected trajectories.Our benchmark also includes fine-grained decomposed substeps, dynamic scene variations, and time-segment annotations, all of which are critical factors in long-horizon tasks yet currently missing in existing benchmarks.

| Benchmarks | Train (var) | Test (var) | Long-horizon | LLM-Gen Tasks | Human Traj. | Dynamic | Time-Anno. | FG. Decomp. |
|---|---|---|---|---|---|---|---|---|
| RLBench [1] | 100 (323) | 100 (323) | ✗ | ✗ | ✗ | ✗ | ✗ | ✗ |
| VLMBench [2] | 8 (233) | 8 (374) | ✗ | ✗ | ✗ | ✗ | ✗ | ✗ |
| ARNOLD [3] | 8 (3,571) | 8 (800) | ✗ | ✗ | ✗ | ✗ | ✗ | ✗ |
| ALFRED [4] | 7 (21,023) | 7 (1,529) | ✓ | ✗ | ✓ | ✗ | ✗ | ✗ |
| Calvin [5] | 34 | 34 (1,000) | ✓ | ✗ | ✓ | ✗ | ✗ | ✗ |
| RoboCasa [6] | 100 | 100 | ✓ | ✓ | ✗ | ✗ | ✗ | ✗ |
| Libero-Long [7] | 10 (500) | 10 (500) | ✓ | ✗ | ✓ | ✗ | ✗ | ✗ |
| VLABench [8] | 100 | 100 | ✓ | ✗ | ✗ | ✗ | ✗ | ✗ |
| **RoboCerebra** | **1,000 (100,000)** | **60 (600)** | ✓ | ✓ | ✓ | ✓ | ✓ | ✓ |

structured System 1–System 2 interaction. The dataset is constructed via a top-down pipeline, where GPT generates task instructions and decomposes them into subtask sequences. Human operators execute the subtasks in simulation, yielding high-quality trajectories with dynamic object variations. Compared to prior benchmarks, RoboCerebra features significantly longer action sequences and denser annotations. We further benchmark state-of-the-art VLMs as System 2 modules and analyze their performance across key cognitive dimensions, advancing the development of more capable and generalizable robotic planners.

# 1 Introduction

Recent advances in vision-language models (VLMs) [9, 10, 11, 12] have introduced new capabilities for robotic manipulation [13, 14]. Departing from conventional control paradigms, recent research [15, 16] has increasingly explored the use of foundation models to enable more generalizable, instruction-conditioned robotic behavior. By integrating VLMs, robotic systems gain enhanced competence in grounding natural language commands within complex visual contexts, thereby improving adaptability across diverse and unstructured environments. A prevalent paradigm instantiates VLMs as fast-reactive System 1 modules—vision-language-action (VLA) models [14, 17, 13] that function as reactive policies, mapping multimodal inputs directly to low-level control signals. While effective for real-time execution, this usage fails to fully exploit the models' strengths in semantic abstraction, relational understanding, and contextual reasoning. These capabilities are fundamentally aligned with slow-thinking System 2 processes [18, 19], such as long-horizon planning and subgoal decomposition. To unlock the full potential of VLMs in robotics, it is imperative to move beyond reactive policy deployment and utilize these models as deliberative planners within hierarchical frameworks.

In line with this vision, recent benchmarks [7, 6, 5, 4] have extended robotic tasks from single-step instructions to multi-step procedures. However, they remain limited in temporal scale and structural complexity, typically involving only 2 to 5 sub-tasks and fewer than 500 action steps [8, 6]. While these benchmarks move beyond early reactive settings, they fall short of capturing real-world demands such as hierarchical goal decomposition, temporal abstraction, and adaptive planning. As a result, the System 2 capabilities of VLMs—particularly high-level reasoning and long-horizon planning—remain underexplored. Addressing this gap requires benchmarks with extended horizons, diverse subgoals, and complex reasoning in dynamic, partially observable environments.

To address these limitations and enable a comprehensive evaluation of System 2 capabilities, we present *RoboCerebra*, a novel benchmark designed to assess long-horizon planning and high-level reasoning in robotic manipulation. RoboCerebra includes: (1) a large-scale manipulation dataset featuring extended task horizons and dynamically evolving environments that better reflect real-world complexity; (2) a baseline framework that integrates System 2–System 1 coordination for hierarchical policy execution; and (3) an evaluation protocol tailored to isolate and measure System 2 performance.

The dataset is constructed in simulation to support long-horizon task evaluation. Since the focus is on high-level reasoning rather than low-level control, the sim-to-real gap is less critical. Simulation further offers scalability and reproducibility, enabling systematic benchmarking. We design an

efficient and high-quality top-down data generation pipeline: GPT [20] is prompted to generate high-level task instructions conditioned on environment context and to decompose them into coherent subtask sequences. To encourage deeper temporal dependencies and complex subgoal structures, we craft prompt strategies that promote long-horizon compositionality. Human operators execute these subtasks in simulation to collect demonstration trajectories, with dynamic object variations introduced to increase scene complexity and semantic diversity. Compared to prior datasets, RoboCerebra provides significantly longer action sequences (6×) and denser subtask annotations, forming a more rigorous testbed for evaluating System 2 reasoning.

To demonstrate the utility of RoboCerebra, we develop a Hierarchical Planning and Execution (HPE) Framework composed of a high-level VLM planner and a low-level VLA controller. The VLM generates structured multi-step plans from low-frequency observations and stores them in a memory bank, while the VLA executes fine-grained actions using high-frequency visual inputs. During training, the VLM learns temporal grounding from video demonstrations, and the VLA acquires visuomotor skills from paired visual inputs and primitive actions. At inference, the VLM updates the plan and memory, and the VLA executes actions accordingly. This hierarchical design combines semantic reasoning with precise control, enabling robust execution in dynamic, long-horizon tasks.

To systematically assess System 2 capabilities, we design an evaluation pipeline that targets key cognitive functions in robotic manipulation. We first train a VLA model at the subtask level as a fixed System 1 controller. Given this shared System 1, we evaluate three critical dimensions of System 2 reasoning through structured System 1–System 2 interaction: (1) planning, the ability to decompose high-level goals into subtask sequences; (2) reflection, the ability to assess task completion status; and (3) memory, the ability to retain and utilize long-term context. We conduct a comprehensive evaluation of state-of-the-art VLMs, including GPT-4o [11], Qwen2.5-VL [21], and LLaVA-Next-Video [22], as System 2 modules and analyze their performance across these dimensions in detail.

## 2   Related Work

**Robotic Manipulation Benchmarks.** Early robotic manipulation benchmarks [1, 2, 3, 7] primarily focus on single-step tasks designed to evaluate low-level control capabilities. With the emergence of more capable simulators and large language models, recent efforts [5, 6, 8] have introduced compositional tasks involving language-conditioned multi-step instructions. However, these benchmarks typically involve fewer than 500 action steps and lack dynamic variations or memory requirements, limiting their ability to reflect real-world task complexity. In contrast, *RoboCerebra* significantly extends action sequence lengths (Fig.1(c)), and explicitly incorporates memory-dependent execution and dynamic scene changes, enabling more comprehensive evaluation of long-horizon reasoning.

**Vision-Language-Action (VLA) Models.** Recent advances in VLMs [9, 10, 12, 11, 23, 24, 25, 26] and large-scale manipulation datasets [27, 28] have driven the development of VLA models [13, 29] that translate language instructions into executable robotic actions. While these models have shown strong performance in short-horizon tasks, their precision is often constrained by discrete action representations. To address this, recent works [30, 31, 32] propose using diffusion models to represent continuous action spaces, improving expressiveness and control fidelity. Nonetheless, existing approaches remain limited in their ability to generalize to long-horizon scenarios. Most focus on low-level policy execution or short-term planning, and struggle to maintain coherence across extended temporal contexts. In this work, we adopt OpenVLA as the low-level controller, and focus on how high-level reasoning via hierarchical System 2 planning can enhance performance in long-horizon manipulation tasks.

## 3   RoboCerebra

In this section, we present *RoboCerebra*, a novel benchmark for evaluating System 2 capabilities in long-horizon robotic manipulation, unified under the vision-language model (VLM) paradigm. *RoboCerebra* comprises three core components: (1) a task suite designed to capture phenomena that naturally unfold over extended temporal horizons, (2) a diverse dataset annotated along multiple dimensions, and (3) a multi-faceted evaluation protocol. Together, these components provide a comprehensive framework for assessing the planning, reasoning, and memory capabilities of VLMs in complex, temporally extended robotic settings.

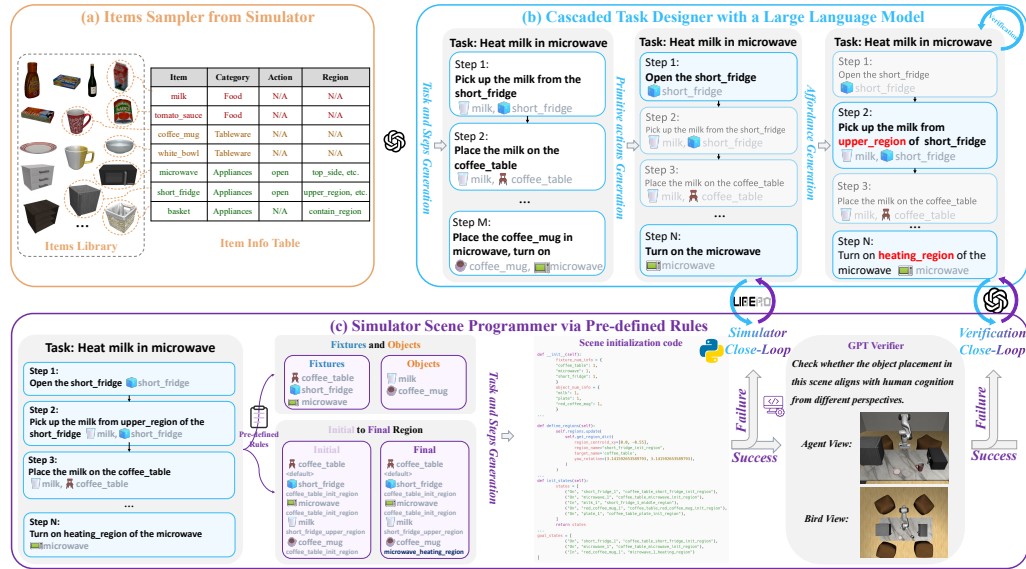

Figure 2: **Task generation pipeline in *RoboCerebra*.** (a) Objects are randomly sampled from Libero's item library and converted into structured representations based on their categories and attributes. (b) The structured data is fed into an LLM to generate high-level task descriptions, which are hierarchically decomposed into low-level substeps. (c) The resulting task plan is parsed into executable simulator code via rule-based transformations. The generated scene is then validated through a closed-loop process involving symbolic checks and vision-language consistency via VLMs.

## 3.1 Task setting

Designing a benchmark for long-horizon robotic manipulation requires addressing challenges absent in short-horizon settings. Beyond decomposing tasks into discrete steps, agents must reason over extended temporal dependencies, operate under partial observability, and adapt to dynamically changing environments. As shown in Fig. 1(b), this entails maintaining memory (e.g., recalling explored cabinet compartments), inferring hidden states (e.g., remembering items placed in closed containers), updating beliefs as the world evolves (e.g., object shifts), and recovering from disruptions (e.g., dropped objects). To capture these complexities, we define six representative sub-task types: *Ideal* — baseline tasks in static, fully observable settings; *Memory Exploration* — requiring active exploration to form internal representations; *Memory Execution* — requiring memory retrieval for goal completion; *Random Disturbance* — introducing unexpected environmental changes; *Observation Mismatching* — requiring robustness to plan-perception misalignment; *Mix* — combining memory and dynamic factors for continual re-planning under uncertainty.

## 3.2 Dataset construction pipeline

We develop a modular pipeline to construct structured, executable tasks for long-horizon robotic manipulation. Built on the Libero simulation platform [7], our pipeline comprises three key stages: (1) *cascaded task generation*, which uses GPT to synthesize high-level tasks and decompose them into subtask sequences; (2) *scene initialization and verification*, which instantiates tasks into physically and semantically valid simulator scenes; and (3) *human demonstration and annotation*, where operators execute subtasks and provide fine-grained temporal labels. This pipeline enables scalable, high-quality dataset construction for evaluating System 2 reasoning in dynamic environments.

**Cascaded Task Generation.** This phase aims to automatically construct structured and executable tasks based on sampled object configurations from the simulator. Given a set of items (Fig. 2(a)), we first convert each object into a structured representation capturing its category, functional affordances, and spatial context. These representations are then used to prompt GPT-o3-mini[20] to generate diverse high-level task descriptions (e.g., "Heat milk in the microwave"), which are subsequently decomposed into coherent step-by-step subtask instructions (Fig. 2(b)). To ensure temporal consistency and physical plausibility, we incorporate affordance-aware and spatially grounded reasoning into the prompting process. The model is guided to validate preconditions, postconditions, and object

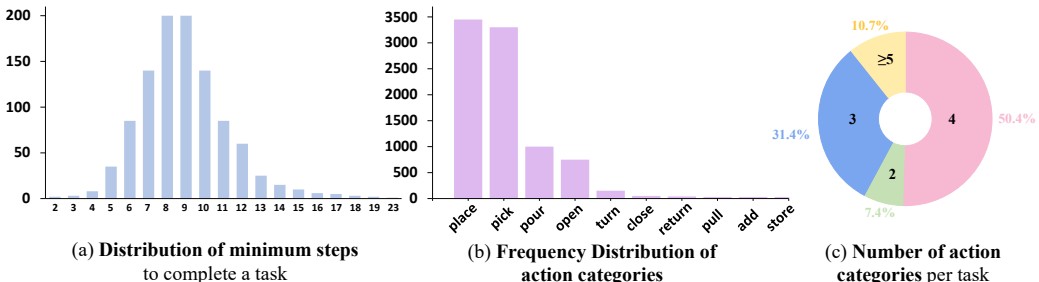

(a) **Distribution of minimum steps** to complete a task

(b) **Frequency Distribution of action categories**

(c) **Number of action categories** per task

Figure 3: **The statistical analysis of our *RoboCerebra* dataset.** . (a) Distribution of minimum steps per task, highlighting its long-horizon nature. (b) Frequency of action categories, with dominant primitives (*place*, *pick*, *pour*) and rare fine-grained actions. (c) Number of action categories per task, showing high compositional diversity—over 10% of tasks involve five or more action types.

interactions across steps, preventing logically invalid or physically infeasible plans. This cascaded generation pipeline enables scalable task creation that is both semantically meaningful and executable, supporting a wide range of object combinations and manipulation goals. The resulting task-subtask pairs serve as the foundation for downstream data collection and learning.

**Scene Initialization and Verification Execution.** This stage aims to translate the structured task instructions into executable scenes within the simulator. Given the cascaded task plans, each action step is parsed into a set of spatial and relational constraints (e.g., placing `milk` from `short_fridge_upper_region` to `coffee_table_top`). We maintain a registry of fixtures and movable objects, and apply rule-based mappings to convert each step into corresponding initial and target object placements (Fig. 2(c)). These placements are then compiled into simulator-executable code to construct the full scene. To ensure that the generated environments are both functionally correct and semantically plausible, we perform two levels of verification. First, a symbolic simulator loop validates the consistency of object states and relational constraints across steps. Second, a vision-language verification loop leverages GPT-4o to evaluate spatial plausibility from multi-view RGB-D renderings, ensuring alignment with human commonsense understanding. This dual-loop validation process ensures that the instantiated scenes are physically realizable and faithfully reflect the intended task semantics.

Table 2: Estimated time for automated and human-in-the-loop stages in our data pipeline. *Gen.*: Cascaded Task Generation, *Program*: Scene Initialization & Verification Programming, *Anno.*: Human Annotation, *Check*: Human Check.

| Stage | Gen. | Program | Anno. | Check |
|---|---|---|---|---|
| **Task** | Prompt | Rule Def. | Traj. & Time | Check |
| **Time** | 20 hrs | 30 hrs | 400 hrs | 200 hrs |

**Human Demonstration and Annotation.** Despite extensive automation, high-quality demonstrations remain essential for learning grounded, long-horizon manipulation. We employ human operators to perform task instructions in simulation, generating diverse and realistic action trajectories. Each trajectory is annotated with fine-grained subtask boundaries to align actions with specific task steps. This subtask-level annotation enables precise temporal segmentation, supports diverse execution styles, and ensures comprehensive coverage across task variants and lengths. It further facilitates grounding of language instructions to motion segments, enabling the training of temporally-aware and instruction-conditioned policies. To maintain annotation quality at scale, we allocate significant human effort: 400 hours for trajectory and time annotation, alongside 200 hours dedicated to a separate verification phase, as outlined in Tab. 2. This rigorous process ensures the consistency, completeness, and accuracy of labeled plans and state transitions.

### 3.3 Data Analysis

RoboCerebra contains 1,000 human-annotated trajectories spanning 100 task variants, each designed to reflect long-horizon and compositional manipulation scenarios (Fig. 3). The tasks cover a wide range of household activities, including preparing drinks, organizing groceries, tidying up, and setting tables. Each task instance consists of 2 to over 20 atomic steps (mean: 9.1), yielding over 10,000

step-level segments with fine-grained temporal annotations. The dataset captures diverse execution patterns across varying spatial and temporal contexts (Fig. 3(a)). To characterize action diversity, we define 12 distinct action types. Common primitives such as *place*, *pick*, and *pour* dominate, while lower-frequency actions like *turn*, *return*, and *store* highlight the fine-grained control required in realistic tasks (Fig.3(b)). On average, each task involves 3.5 action categories, and over 10% of tasks require five or more, indicating high compositional complexity (Fig. 3(c)). A defining feature of RoboCerebra is its extended temporal scale. As shown in Fig. 1(c), the average trajectory length reaches 2,972.4 simulation steps—about 6× longer than existing long-horizon manipulation datasets. This temporal richness supports the study of memory-based control, subgoal abstraction, and planning under long-term dependencies. Moreover, the dataset maintains a broad distribution over trajectory lengths and task types, facilitating evaluation across a wide spectrum of planning challenges.

### 3.4 Multi-dimensional Evaluation

We evaluate system performance across a benchmark of $N$ long-horizon manipulation tasks. Each task $i$ is defined by a sequence of $K_i$ key object state transitions, denoted as $s_i^{(1)}, s_i^{(2)}, \ldots, s_i^{(K_i)}$, which constitute the minimal sufficient conditions for successful task completion. These transitions are automatically verified using a simulator-internal predicate function $\psi(s)$, which returns True if the target condition holds in state $s$.

While binary task success is a standard metric, it alone fails to capture the broader cognitive demands of long-horizon manipulation. We therefore propose a multi-dimensional evaluation protocol grounded in four complementary metrics: (1) *task success*, (2) *planning accuracy*, (3) *planning efficiency*, and (4) *action completion accuracy*. Each metric targets a distinct aspect of long-horizon reasoning, enabling a more comprehensive understanding of system behavior.

To evaluate planning efficiency, we compare the predicted high-level plan $\pi_i^{\text{pred}}$ generated by a large language model (LLM) against a human-annotated ground-truth plan $\pi_i^{\text{GT}}$ using exact sequence matching. The actual symbolic execution trace is denoted as $\mathcal{A}_i = [a_1, a_2, \ldots, a_T]$, where $a_t$ represents a discrete symbolic action at step $t$.

To assess perceptual alignment and semantic interpretability, we introduce a VideoQA benchmark comprising $M$ human-written binary questions $q_j{}_{j=1}^M$. Each question is evaluated using a verification function $\delta(q_j)$, which returns 1 if the correct answer is inferred from the execution, and 0 otherwise. We define the following evaluation metrics:

**Task Success Rate** ($SR$)**:** Measures whether the agent achieves the intended object state transitions:

$$\text{SR}_i = \frac{1}{K_i} \sum_{k=1}^{K_i} \mathbf{1}\left[\psi\left(s_i^{(k)}\right)\right] \tag{1}$$

**Average Plan Match Accuracy** ($\text{Acc}_P$)**:** Measures the average agreement between predicted and ground-truth high-level plans:

$$\text{Acc}_P = \frac{1}{N} \sum_{i=1}^{N} \mathbf{1}\left[\pi_i^{\text{pred}} = \pi_i^{\text{GT}}\right] \tag{2}$$

**Plan Efficiency** ($\eta$)**:** Measures the efficiency of symbolic planning by dividing the success rate by the average plan length:

$$\eta = \frac{\text{SR}}{\text{Len}} = \frac{\text{SR}}{\frac{1}{N} \sum_{i=1}^{N} |\mathcal{A}_i|} \tag{3}$$

**Action Completion Accuracy** ($Acc_C$)**:** ssesses semantic interpretability via the QA benchmark, a metric closely associated with the model's reflection ability:

$$\text{Acc}_C = \frac{1}{M} \sum_{j=1}^{M} \mathbf{1}\left[\delta(q_j)\right] \tag{4}$$

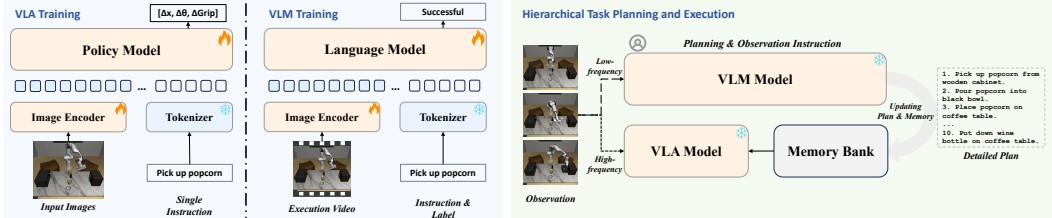

Figure 4: **Overview of our HPE framework.** *Left:* VLA model training uses paired images and single-step instructions to optimize a visual token policy. VLM training uses execution videos with success-labeled instructions for temporal grounding. *Right:* During execution, the VLM processes low-frequency observations to update low-level plans stored in memory bank, while the VLA consumes high-frequency observations to execute fine-grained actions based on the detailed plan.

# 4   Hierarchical Planning and Execution Framework

To fully leverage the complementary strengths of high-level reasoning and low-level control, we propose a Hierarchical Planning and Execution (HPE) Framework. As shown in Fig. 4, the system consists of two components: a VLM that operates on low-frequency egocentric observations to generate and update step-level subgoals, and a VLA that consumes high-frequency inputs to execute fine-grained actions conditioned on subgoals. A shared memory bank mediates communication between the two modules, enabling dynamic coordination and closed-loop execution across the task.

## 4.1   Training Procedure

To support effective coordination within the hierarchical system, we adopt a two-stage supervised fine-tuning paradigm, providing targeted supervision for both modules. This strategy equips the VLA with low-level control capabilities and the VLM with high-level reasoning and progress monitoring skills. In the first stage, we train the VLA to execute fine-grained actions based on egocentric observations and step-level instructions. Training data is derived from long-horizon demonstrations, where each (image, instruction) pair is paired with the corresponding robot action. Following OpenVLA [13], continuous actions are discretized into token sequences, and the model is trained via next-token prediction. This enables the VLA to acquire reusable visuomotor primitives, forming a reliable foundation for generalizable low-level control. In the second stage, we train the VLM to interpret video-instruction pairs and assess task progress. To enable this, we construct a dataset of videos annotated with step-level instructions, including both successful and incomplete executions. Using contrastive supervision, the model learns to associate visual sequences with instruction completion status. This allows the VLM to perform progress-aware planning and re-planning based on real-time visual feedback. By explicitly modeling execution status, the VLM enhances robustness and facilitates tighter coordination with the low-level controller during long-horizon tasks.

## 4.2   Hierarchical Task Planning and Execution

At inference time, the VLM parses a high-level task instruction into a sequence of step-level subgoals, which are stored in a memory bank. The VLA continuously queries the active subgoal and executes corresponding low-level actions based on high-frequency visual observations. Concurrently, the VLM periodically attends to recent observations to monitor execution progress. Upon detecting subgoal completion or deviation, it updates the memory with the next subgoal or issues a refined instruction. This closed-loop coordination preserves temporal abstraction while ensuring reactive control, enabling robust and interpretable performance in long-horizon tasks.

# 5   Experiments

## 5.1   Experimental Settings

**System 1 Models.** To adapt the VLA model to our long-horizon domain, we sample 100 task instances from the RoboCerebra training set and decompose each into single-step sequences based

Table 3: Performance comparison across six sub-tasks. * indicates models fine-tuned on our data. Average success rates (%) are reported over 10 rollouts for each method on Random Disturbance (Ran.), Observation Mismatching (Obs.), Memory Exploration (Exp.), Memory Execution (Exe.), Mix, and Ideal tasks.

| Method | Avg | Dynamic | | Memory | | Mix | Ideal |
|---|---|---|---|---|---|---|---|
| | | Ran. | Obs. | Exp. | Exe. | | |
| OpenVLA-Libero100 | 2.00 | 4.59 | 1.35 | 0.18 | 1.86 | 0.00 | 4.05 |
| OpenVLA* | 4.57 | 7.84 | 8.65 | 1.06 | 2.06 | 0.00 | 7.84 |
| Planner+OpenVLA* | 16.04 | 18.63 | **19.45** | 8.04 | 16.69 | 11.48 | **21.92** |
| Hierarchical Framework | **16.55** | 18.63 | 19.18 | **9.06** | **17.83** | **13.21** | 21.10 |

on temporal annotations. We fine-tune OpenVLA [13] on this dataset. The model is trained for 200K steps with a global batch size of 64, an initial learning rate of 5e-5 (decayed after 100K steps), and an input resolution of 256×256. Training and evaluation are conducted on 8 NVIDIA A100 GPUs.

**System 2 Models.** We evaluate the reasoning capabilities of different System 2 models across multiple settings. Specifically, we consider three categories: (1) pre-trained VLMs (GPT-4o [11], Qwen2.5-VL [21], and LLaVA-Next-Video [22]), (2) blind LLMs (visual input disabled to isolate language-based reasoning), and (3) a supervised fine-tuned VLM trained on our video-instruction dataset, labeled for success and failure at the subtask level. Each System 2 model operates as a high-level controller that generates step-wise instructions executed by a fixed System 1 policy. Thus, task success primarily reflects the reasoning capability of the System 2 module.

**Baselines.** We further study System 2's role across distinct cognitive functions: *planning*, *observation*, and *memory*. Based on these axes, we implement multiple baselines. For example, a Planner baseline generates the entire subgoal sequence from the initial instruction without further perception or feedback. To better exploit the capacity of VLMs, we implement a **Hierarchical Framework** (Fig. 4) where System 2 dynamically monitors the environment, updates subgoals, and interfaces with a memory module for long-horizon planning.

**Evaluation Protocol.** We evaluate each method over 600 rollouts (60 tasks × 10 trials). For fair comparison across planning models, we define a set of anchor points that determine when System 1 transitions between subgoals. These anchor-aligned transitions decouple step-switching from the model, allowing consistent temporal granularity across models. Detailed metrics are in Sec. 3.4.

## 5.2 Main Results

**System 1 struggles in long-horizon tasks.** As is shown in Tab. 3, the OpenVLA [13] model exhibits substantial limitations when applied to long-horizon manipulation. In the *Ideal* setting—designed to isolate long-horizon reasoning under static and fully observable conditions—it achieves only a success rate of $4.05\%$, highlighting its difficulty in executing extended instruction sequences. Although supervised fine-tuning slightly improves performance to $7.84\%$, it remains far below the $21.10\%$ achieved by our Hierarchical Framework. These results suggest that OpenVLA's architecture lacks the capacity to maintain instruction fidelity across long temporal spans. Furthermore, in the *Mix* setting, which introduces both memory demands and dynamic scene variations, the fine-tuned OpenVLA fails completely with a success rate of $0.00\%$, reinforcing its inability to handle temporally extended dependencies and partial observability in complex environments.

**System 2 enhances System 1 in more complex tasks.** In the *Mix* setting, where long-term memory and adaptive planning are essential, both Planner+OpenVLA and HPE show notable gains, achieving success rates of $11.48\%$ and $13.21\%$, respectively. These results indicate that incorporating System 2 planning improves performance in memory-intensive scenarios, and that iterative reasoning through the VLM, as enabled by the hierarchical approach, offers further benefits. However, in the *Ideal* setting, the Hierarchical Framework performs slightly below Planner+OpenVLA, with a $0.82\%$ drop in success rate. This suggests that in simpler, fully observable tasks, the additional reasoning overhead may introduce unnecessary complexity, potentially leading to suboptimal decisions.

Table 4: Ablation Study on Different Planner Model (%). Blind denotes models without visual input, while GT-plan denotes following the ground-truth plan directly. **Bold** numbers indicate the best performance, and underlined numbers indicate the second-best performance.

| Planner Model | *Avg* | *Dynamic* | | *Memory* | | *Mix* | *Ideal* |
|---|---|---|---|---|---|---|---|
| | | Ran. | Obs. | Exp. | Exe. | | |
| GT-plan | 25.16 | 26.85 | 30.68 | 19.47 | 23.48 | 19.26 | 31.23 |
| Qwen2.5-VL-Blind | 11.87 | 18.90 | 12.88 | 7.02 | 10.87 | 2.55 | 18.90 |
| LLaVA-Next-Blind | 8.00 | 13.97 | 12.33 | 3.54 | 3.54 | 0.37 | 14.25 |
| GPT-4o-Blind | 15.10 | **20.00** | 17.03 | 7.02 | 16.09 | 10.48 | 20.00 |
| Qwen2.5-VL | 11.19 | 14.25 | 14.25 | 2.63 | 12.61 | 6.67 | 16.71 |
| LLaVA-Next-Video | 11.37 | 16.71 | 16.16 | 1.07 | 10.87 | 3.70 | 19.73 |
| GPT-4o | **16.04** | 18.63 | **19.45** | **8.04** | **16.69** | **11.48** | **21.92** |

## 5.3 Ablation on Planner

As shown in Tab. 4, we conducted ablation studies under different Planner Models. In this set of experiments, GPT-4o achieved an average success rate of 16.04%, surpassing other VLMs by more than 4%. Moreover, even with visual inputs removed, GPT-4o still maintained relatively stable performance, indicating that the reasoning capabilities of VLMs can significantly contribute to solving long-horizon tasks. However, there remains a performance gap of over 9% between GPT-4o and the GT-plan, suggesting that the lack of interaction with the environment and the domain gap in visual understanding may both lead to performance degradation in the system.

Table 5: Evaluation of System 2 capabilities from multiple perspectives, including Average Planning Accuracy ($Acc_P$), Observation Judgment ($Acc_C$), Success Rate ($SR$), Average Plan Length ($Len$) and Plan Efficiency ($\eta$).

| Model | $Acc_P \uparrow$ | $Acc_C \uparrow$ | $SR \uparrow$ | $Len \downarrow$ | $\eta \uparrow$ |
|---|---|---|---|---|---|
| GPT-4o | **68.33** | 32.66 | **16.04** | 10.67 | **1.50** |
| GPT-4o-Blind | 61.37 | 0.00 | 15.10 | 10.73 | 1.41 |
| LLaVA-Next-Video-7B | 40.00 | 37.19 | 11.37 | 8.33 | 1.36 |
| Qwen2.5-VL-7B | 44.67 | 47.74 | 11.19 | 8.30 | 1.34 |
| Qwen2.5-VL-7B-SFT | 30.00 | **66.83** | 9.33 | **6.95** | 1.32 |

## 5.4 Evaluation on System 2

As shown in Tab. 5, we evaluate System 2 by switching among different categories of VLMs across metrics. GPT-4o achieves the best performance in planning accuracy, task success rate, and plan efficiency, demonstrating that the reasoning capabilities of a System 2 model can directly contribute to the planning and execution of long-horizon tasks. Although GPT-4o performs relatively poorly in terms of simulator observation, it still outperforms Qwen2.5-VL-7B-SFT by more than 6.5% in task success rate. This suggests that environmental observation has not yet played a decisive role in the completion of long-horizon tasks. $Acc_C$ is mainly used to evaluate the model's capability for reflection, and Qwen2.5-VL significantly improves its reflection ability in the simulation environment after fine-tuning.

## 6 Conclusion

In this work, we introduce RoboCerebra, a benchmark for evaluating System 2 capabilities in long-horizon robotic manipulation. By focusing on multi-step tasks in dynamic environments, it addresses limitations of prior reactive benchmarks. Through top-down task generation and a hierarchical planning framework, RoboCerebra enables systematic evaluation of planning, reflection, and memory. Experiments with state-of-the-art VLMs highlight varying reasoning capabilities, underscoring the need for more robust, temporally grounded decision-making in robotics.

**Limitations.** This work offers an initial exploration of System 2 capabilities via a plan-memory-based hierarchical framework. However, the interaction between System 1 and System 2 remains limited. Future work could support richer bidirectional communication, enabling finer-grained, interpretable feedback. The evaluation protocol may also be extended with execution-level signals such as subtask ordering and failure recovery. Lastly, deploying the benchmark in real-world settings would introduce additional challenges and further validate long-horizon reasoning under realistic conditions.

# 7 Acknowledgements

This research is supported in part by National Key R&D Program of China (2022ZD0115502), Ningbo Science and Technology Innovation 2025 Major Project (2025Z034), National Natural Science Foundation of China (NO. 62461160308, U23B2010), "Pioneer" and "Leading Goose" R&D Program of Zhejiang (No. 2024C01161). In addition, this research was supported in part by NUS Start-up Grant A-0010106-00-00.

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

# A Appendix

## A.1 Detailed Task Suite

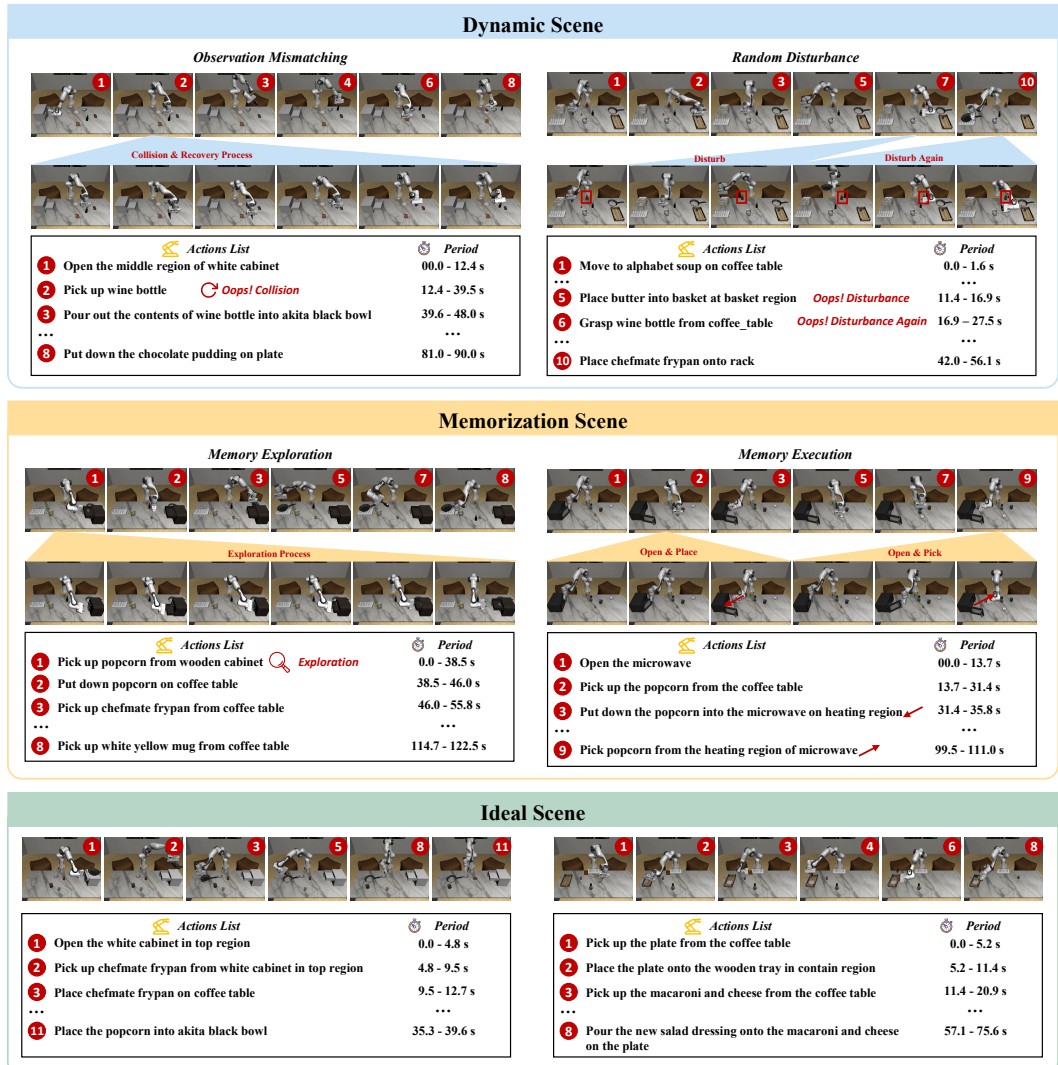

Figure 5: **Example from *RoboCerebra* for different tasks**.

Fig. 5 illustrates concrete examples of our defined sub-task categories using the *RoboCerebra* benchmark. These scenarios are designed to evaluate the capabilities of System 2 models in memory retention, adaptive planning, and disturbance recovery within long-horizon manipulation settings. We group these into three broader categories for analysis:

(1) Dynamic Scene includes Observation Mismatching and Random Disturbance sub-tasks. These emphasize robustness to environmental inconsistencies and unexpected changes. For example, Systems must recover from collisions or update plans after items are displaced mid-task.

(2) Memorization Scene covers both Memory Exploration and Memory Execution. In the exploration phase, VLMs actively probe the environment to build internal representations (e.g., checking cabinet contents). During execution, perceptual cues are removed and they rely on memory to complete the task correctly.

(3) Ideal Scene serves as a static, fully observable control condition. It reflects performance in the absence of memory or disturbance constraints and establishes a baseline for comparison.

Each scenario is visualized as a sequence of image snapshots, with numbered action steps, annotated time segments, and accompanying descriptions of the behavior of System 2. These task compositions holistically assess core abilities such as long-horizon memory, temporal reasoning, causal inference, and goal-directed manipulation in evolving environments.

## A.2 Study on Different VLA models

Table 6: Performance comparison of VLMs under OpenVLA and $\pi$0-fast. Results show that the current VLM maintains comparable performance across both settings, indicating that its System 2 capability is independent of the specific VLA architecture.

| Method | Avg | Ran. | Obs. | Exp. | Exe. | Mix | Ideal |
|---|---|---|---|---|---|---|---|
| Qwen2.5-LM-OpenVLA | 11.87 | 18.90 | 12.88 | 7.02 | 10.87 | 2.55 | 18.90 |
| Qwen2.5-LM-$\pi$0-fast | 11.47 | 16.32 | 11.94 | 9.92 | 8.27 | 4.73 | 17.63 |
| LLaVA-N-Blind-OpenVLA | 8.00 | 13.97 | 12.33 | 3.54 | 3.54 | 0.37 | 14.25 |
| LLaVA-N-Blind-$\pi$0-fast | 5.99 | 7.89 | 8.06 | 5.89 | 3.85 | 1.82 | 8.42 |
| GPT-4o-Blind-OpenVLA | 15.10 | 20.00 | 17.03 | 7.02 | 16.09 | 10.48 | 20.00 |
| GPT-4o-Blind-$\pi$0-fast | 13.63 | 14.47 | 15.82 | 12.27 | 12.69 | 11.27 | 15.26 |
| Qwen2.5-VL-OpenVLA | 11.19 | 14.25 | 14.25 | 2.63 | 12.61 | 6.67 | 16.71 |
| Qwen2.5-VL-$\pi$0-fast | 13.19 | 20.79 | 15.45 | 8.24 | 6.15 | 6.91 | 21.58 |
| LLaVA-N-video-OpenVLA | 11.37 | 16.71 | 16.16 | 1.07 | 10.87 | 3.70 | 19.73 |
| LLaVA-N-video-$\pi$0-fast | 8.79 | 12.11 | 12.12 | 7.73 | 4.81 | 2.55 | 13.42 |
| GPT-4o+OpenVLA | 16.04 | 18.63 | 19.45 | 8.04 | 16.69 | 11.48 | 21.92 |
| GPT-4o+$\pi$0-fast | 15.15 | 18.95 | 20.00 | 10.59 | 11.73 | 10.18 | 19.47 |
| GT-plan-OpenVLA | 25.16 | 26.85 | 30.68 | 19.47 | 23.48 | 19.26 | 31.23 |
| GT-plan-$\pi$0-fast | 23.04 | 23.68 | 26.36 | 18.15 | 16.92 | 26.55 | 26.58 |

To verify whether the System 2 reasoning capability of the current VLM depends on a specific VLA architecture, we conducted additional experiments comparing two System 1 modules — OpenVLA and $\pi$0-fast (implemented following the openpi repository).

As shown in Table 6, the model exhibits comparable performance under both configurations across all evaluated metrics. This consistency suggests that the higher-level cognitive and planning abilities of the VLM function independently of the underlying low-level control mechanism. Although slight variations can be observed in metrics such as Execution and Exploration, the overall performance trend remains stable. These results demonstrate that the reasoning and decision-making processes of the VLM are robust and transferable across different embodied control implementations, confirming the generality and architecture-independence of its System 2 module.

## A.3 Study on Different Planners

Table 7: Comparison of specialized embodied VLMs with general-purpose models. Specialized VLMs show stronger performance on Embodied-QA benchmarks and complex scenarios (Exp. and Mix), though their abilities have not fully generalized to Embodied Planning.

| Method | Para. | Avg | Ran. | Obs. | Exp. | Exe. | Mix | Ideal |
|---|---|---|---|---|---|---|---|---|
| LLaVA-N-Blind | 7B | 8.00 | 13.97 | 12.33 | 3.54 | 3.54 | 0.37 | 14.25 |
| Cosmos-Reason1 | 7B | 8.41 | 7.63 | 10.45 | 5.55 | 7.31 | 8.73 | 10.79 |
| VeBrain | 8B | 9.41 | 12.89 | 12.35 | 7.06 | 3.65 | 4.21 | 16.32 |
| Qwen2.5-VL | 7B | 11.19 | 14.25 | 14.25 | 2.63 | 12.61 | 6.67 | 16.71 |
| LLaVA-N-Video | 7B | 11.37 | 16.71 | 16.16 | 1.07 | 10.87 | 3.70 | 19.73 |
| RoboBrain-2.0 | 7B | 11.40 | 12.11 | 12.06 | 9.92 | 10.96 | 7.27 | 16.05 |
| GPT-4o | - | 16.04 | 18.63 | 19.45 | 8.04 | 16.69 | 11.48 | 21.92 |

As shown in Table 7, VLMs specifically designed for embodied tasks—such as RoboBrain-2.0, Cosmos-Reason1, and VeBrain—generally outperform general-purpose VLMs on the Embodied-QA benchmark. These specialized models also exhibit superior adaptability in our constructed complex tasks, particularly in challenging scenarios such as Memory Exploration and Mix, which require long-horizon reasoning and dynamic decision-making.

However, this specialization has not yet fully transferred to the domain of Embodied Planning, where limitations remain in generating coherent planning sequences and decomposing actions into fine-grained steps. This observation suggests that, while task-specific VLMs demonstrate stronger contextual understanding and perception-grounded reasoning, their higher-level planning and general reasoning abilities are still developing. These results highlight an important direction for future research on embodied intelligence — achieving deeper integration between perception, reasoning, and long-term planning.

## A.4 Results on Memory Tasks

Table 8: **Evaluation of different VLMs** under the Hierarchical Framework, including Memory Exploration Success Rate ($SR_{Exp.}$), Exploration-only Success Rate ($SR_{Exp.-only}$), Exploration Efficiency ($\eta_{Exp.}$), Memory Execution Success Rate ($SR_{Exe.}$), and Decision Accuracy ($Acc_{Dec.}$).

| VLM | $SR_{Exp.}$ ↑ | $SR_{Exp.-only}$ ↑ | $\eta_{Exp.}$ ↑ | $SR_{Exe.}$ ↑ | $Acc_{Dec.}$ ↑ |
|---|---|---|---|---|---|
| Qwen2.5-VL | 3.54 | 50.0 | 0.17 | 12.39 | 10.0 |
| GPT-4o | **9.06** | **80.0** | **0.32** | **17.83** | **30.0** |

To further evaluate the role of System 2 reasoning in Memory-based manipulation tasks, we design a set of fine-grained experiments to analyze how different reasoning capabilities affect performance under a unified Hierarchical Framework. The detailed memory mechanism is illustrated in Alg. 1 Beyond the overall success rates of the Memory Exploration and Memory Execution tasks, we introduce several intermediate metrics that assess the internal reasoning process:

To evaluate the ability of System 2 to discover the target object during the exploration phase, we utilize the **Exploration-only Success Rate** ($SR_{Exp.-only}$), which measures whether the VLMs successfully locates the object regardless of overall task completion.

To assess the efficiency of object discovery during exploration, we utilize the *Exploration Efficiency* metric ($\eta_{Exp.}$), which jointly considers the correctness and conciseness of the predicted exploration plan. Specifically, we define it based on the normalized overlap between the predicted plan $\pi_G$ and the ground truth plan $\pi_{GT}$ for each task $i$, referred to as the completeness of the plan:

$$Comp_{Exp.} = \frac{|\pi_G \cap \pi_{GT}|}{|\pi_{GT}|} \qquad (5)$$

The exploration efficiency is then computed by normalizing the completeness score by the length of the predicted plan and averaging over all $N$ tasks:

$$\eta_{Exp.} = \frac{1}{N} \sum_{i=1}^{N} \frac{Comp_{Exp.}}{|\pi_G|} \qquad (6)$$

To measure the correctness of high-level decision-making during execution, we utilize the **Decision Accuracy** ($Acc_{Dec.}$), which quantifies the proportion of correct identifications of the target object and appropriate plan selections.

## A.5 Prompt Details of Task Generation

To support structured generation of long-horizon manipulation tasks, we employ a multi-stage prompting pipeline to elicit detailed, consistent, and actionable representations from large language models. Each stage is designed to incrementally refine the task definitions, ensuring coherence, atomicity, and alignment with robotic capabilities. The full set of prompt templates used in our pipeline is shown in Fig. 6–10.

Specifically, Fig. 6 shows the Task and Steps Generation Prompt, which outlines the high-level goal and asks the model to break it down into discrete procedural steps. Following this, the Task and Steps Verification Prompt verifies the semantic and logical validity of these decomposed steps (Fig. 7). Once verified, we proceed with Primitive Actions Generation (Fig. 8), where each high-level step is

**Algorithm 1** Task-Aware Memory Mechanism

---

1: **Input:** Task specification $\mathcal{T}$, current environment state $s$
2: **Output:** Success or Failure
3: Determine whether $\mathcal{T}$ requires memory (e.g., exploration, history tracking)
4: Define goal condition $G$ from $\mathcal{T}$
5: Identify goal-relevant steps $\{s^{(1)}, s^{(2)}, \ldots, s^{(k)}\}$
6: Generate complete sub-plan $\pi_G$ to achieve $G$
7: **for** each step $a_t$ in $\pi_G$ **do**
8:     Execute $a_t$ in environment
9:     **if** $\psi(s_t) = \texttt{True}$ **and** $s_t$ satisfies goal $G$ **then**
10:         **return** Success
11:     **end if**
12: **end for**
13: **return** Failure

---

further grounded into low-level robot-executable primitives. Subsequently, the Affordance Generation Prompt (Fig. 9) augments the plan with relevant object affordances, aiding visual grounding and interaction feasibility. Finally, a Format Verification Prompt (Fig. 10) ensures that the entire structured output adheres to the expected schema, enabling seamless downstream parsing and simulation.

---

**Stage 1: Task and Step Generation Prompt**

You are an imaginative robotic arm scene designer. You need to construct a long-sequence task based on the item list provided below, which includes multiple steps (no more than **8 steps**), with each step being **an action** (such as "open the xxx", "Pick up the xxx from xxx", "open the xxx" etc.). All the items were initially placed on the {workspace} or stored inside other objects.
Note:
1. Except for items that offer executable actions, other items can only be picked up, moved, and put down. You are not allowed to open any food packaging boxes. If you wish to pour out any liquid from the container, simply pour it out—there's no need to open the packaging. The scene does not contain any real liquids.
2. This long sequence of tasks does not necessarily require a complete chain of events.
3. You don't have to use all the items—please prioritize ensuring that the constructed task logic is reasonable.
4. Can_fit is a boolean value, which indicates whether the object can be stored in any storage container.
5. In 'Related object', list and only list all the NAME of objects involved in the step, including the object being operated on and the object being moved.
6. DONOT output any irrelevant replies. The format of the replies is as follows:
   Task: xxx
   Step: xxx
   Related Objects: xxx
   Step: xxx
   Related Objects: xxx
   ....
   The item table is as follows:
   Item   Category   Action   Affordance   Can_fit
   ....

Figure 6: **Task and Steps Generation Prompt.**

---

**Stage 1 Verification: Rationality Check**

Please help me check and correct the previously generated task.

Note:

1. The robot only has **one robotic arm**, it is **fixed to the table**. After picking up an item, it must place it first before picking up another item.
2. DONOT output any irrelevant replies. The format of the replies is as follows:

   Task: xxx

   Step: xxx

   Related Objects: xxx

   Step: xxx

   Related Objects: xxx

   ....

   Here is the task and steps:

   ....

---

Figure 7: **Task and Steps Verification Prompt.**

---

**Stage 2:Primitive Actions Generation**

Please break down each of the following steps into **atomic tasks**, such as "open the xxx", "Pick up the xxx", "Put the xxx" etc.

Note:

1. DONOT output any irrelevant replies. The format of the replies is as follows:

   Task: xxx

   Step: xxx

   Related Objects: xxx

   Step: xxx

   Related Objects: xxx

   ....

   Here is the task and steps:

   ....

---

Figure 8: **Primitive Actions Generation Prompt.**

## A.6 Prompt Details of Hierarchical Framework

To support reasoning over long-horizon manipulation tasks involving memory and dynamic scene understanding, we adopt a hierarchical prompting framework that decomposes planning into semantically modular components. This design enables the VLMs to perform goal decomposition, plan generation, and state tracking in a structured and interpretable manner. Fig. 11 to 13 illustrate the key prompts used in this hierarchical process.

As shown in Fig. 11, the VLM Planner Prompt guides a vision-language model to generate high-level plans based on multimodal inputs, including current visual observations and task descriptions. This prompt leverages the VLM's capability to contextualize semantic goals within visual environments.

Fig. 12 presents the Memory-Related Goal Generation Prompt, which is used to infer intermediate or hidden sub-goals that depend on past interactions or occluded states. This is particularly important in scenarios where successful execution requires recalling previously explored regions or remembering the contents of closed containers.

Finally, the Plan and Memory Updating Prompt (Fig. 13) enables continual re-planning by integrating updated perceptions and internal memory states. This prompt ensures that the VLMs maintains coherence between its execution state and the evolving environment, allowing it to revise intentions and recover from deviations or external disturbances.

**Stage 3 :Affordance Generation**

Please add the region where the object is located in each sentence that describes the step. The region refers to those **listed in the "affordance" column**.
Note:
1. Please keep the format of this response consistent with the input task and step format.
2. The 'region' in the 'affordance' column refers to the region that belongs to the object, not the area where the object is located.
3. Do not modify the sentence that describs the related objects.
4. You can't place 2 object in the same region.
5. DONOT output any irrelevant replies. The format of the replies is as follows:
   Task: xxx
   Step: xxx
   Related Objects: xxx
   Step: xxx
   Related Objects: xxx
   ....
   Here is the task and steps:
   Item   Category   Action   Affordance   Can_fit

Figure 9: **Affordance Generation Prompt.**

## A.7 Case Study on Planning

To better understand the planning capabilities of different vision-language models, we conduct a qualitative comparison across GPT-4o [11] (GPT), Qwen2.5-VL [21] (Qwen), and LLaVA-Next-Video [24] (LLaVA) on complex household manipulation tasks, as illustrated in Fig. 14 and Fig. 15. These examples highlight notable differences in plan completeness, action granularity, and semantic correctness.

GPT consistently generates the most detailed and coherent plans across both tasks. In Fig. 14, GPT not only identifies the correct goal ("placing the plates between the knife and fork") but also incorporates contextual spatial cues and restores objects to their original positions (e.g., placing the wine bottle back to the far-left side), reflecting strong planning fidelity and environmental awareness. Similarly, in Fig. 15, GPT successfully decomposes a long-horizon task into logically ordered steps while preserving semantic consistency across diverse object types and destinations, demonstrating robustness in long-sequence planning.

Qwen produces generally correct but less detailed plans. In both examples, Qwen omits several contextual elements, such as spatial descriptors or placement constraints, which reduces interpretability and precision. Notably, in Fig. 15, it fails to include important object transitions (e.g., placing the mug into the tray), indicating partial task understanding and missing intermediate steps.

LLaVA, in contrast, struggles with both semantic accuracy and action decomposition. It introduces several unnecessary or incorrect steps—such as manipulating objects irrelevant to the goal or reversing object trajectories. For example, it incorrectly places the white-yellow mug into the wooden tray from the wrong starting location and includes object movements not specified in the task. This suggests that LLaVA lacks grounded task representations and often fails to maintain consistency with the initial instructions.

## A.8 Case Study on Sub-Plan

Fig. 16 presents a comparison of sub-task decomposition quality between GPT and Qwen, focusing on the process of locating and retrieving butter from a multi-compartment cabinet. The results highlight distinct differences in the models' ability to reason hierarchically and generate coherent sub-plans.

Please help me check whether there are any formatting issues in the output below.

You are a helpful assistant that converts high-level tasks into detailed, step-by-step physical actions.

For each task, output a series of action steps in the following format:

Steps:

xxxxx (only a sentence of instruction)

Example:

Task: Took a plate from the wooden cabinet, set it on the coffee table, turned on the stove, poured wine from a bottle onto the plate, and placed a red coffee mug next to it.

Steps:

Open the wooden cabinet top region

Pick up the plate from wooden cabinet top region

Place the plate on the coffee table

Turn on the flat stove cook region

Pick up the wine bottle from coffee table

Pour out the wine bottle onto the plate

Place the wine bottle on the coffee table

Pick up the red coffee mug from coffee table

Place the red coffee mug beside the plate on the coffee table

Make sure to:

- Identify **all relevant objects** in the scene

- Choose appropriate locations and relative positions

- The wooden cabinet has top, middle, and bottom regions that can store objects and be opened

- Maintain a logical and efficient order

- Ensure that all objects used in each step must come exclusively from the list of "Objects on the table"

- DONOT output any content except steps

Now complete the following task step-by-step:

Task: {task_input}.

Object on the table: {obj_input}

Steps:

Figure 10: **Format Verification Prompt.**

GPT demonstrates a thorough and methodical decomposition strategy. It systematically explores all cabinet compartments in a top-down order, includes both opening and closing actions, and maintains logical sequencing. This behavior reflects robust sub-task planning and a clear understanding of both spatial structure and task completion integrity. In contrast, Qwen exhibits incomplete and less stable sub-plan generation. It assumes the butter is in the first compartment without exploration and omits intermediate or fallback steps entirely. This leads to a plan that may work in specific instances but lacks generality and reliability in realistic or uncertain environments.

## A.9  Case Study on Memory Tasks

We evaluate models' capabilities to reason about memory-related goals and update task completion status accordingly. As shown in Fig. 17 and 18, we compare GPT and Qwen on two representative settings: memory exploration and memory execution.

In the memory exploration task (Fig. 17), the VLMs need to track visual progress and determine when a goal—retrieving butter from a cabinet—has been fulfilled. GPT demonstrates clear task completion awareness, identifying in the third step that the butter has been found inside the open

Figure 11: **VLM planner Prompt.**

cabinet and marking the goal as completed. This reflects its ability to connect visual observations with memory-dependent goals and dynamically update task status. In contrast, Qwen remains task completion-unaware, repeatedly denying goal fulfillment despite the presence of the butter in view. This suggests difficulty in grounding visual evidence against prior memory constraints.

In the memory execution task (Fig. 18), the goal is to check whether a previously interacted cabinet contains any remaining objects. GPT again shows strong reasoning, correctly concluding the goal is satisfied when the cabinet is observed to be empty. Its output highlights an understanding of absence as valid evidence. On the other hand, Qwen fails to interpret the empty cabinet scene as sufficient for task completion, citing uncertainty and lack of confirmation. This highlights a limitation in negative inference, where models must reason not just over what is visible, but also over what is not.

**Hierarchical Framework: Memory Related Goal Generation**

You are a helpful assistant that can decide whether a task requires memory to finish.

Generate a goal that requires memory to complete — for example, it might involve searching through multiple compartments or drawers to find an object, where the agent needs to remember which areas have already been searched to avoid redundant actions.

Make sure that:
- Identify all relevant objects in the scene
- The wooden cabinet has top, middle, and bottom regions that can store objects and be opened

Example1:
  -Task:
  Take the white bowl from the wooden cabinet, pour in the chocolate pudding and cookies, and put it on top of the wooden cabinet.
  -Output Format:
  MEMORY RELATED GOAL:Take the white bowl in the cabinet.
  -Reason for Output(DO NOT output this part):
  This instruction guides the user to complete a goal-oriented task. To achieve the goal, the user must search through the top, middle, and bottom compartments of the wooden cabinet. Since the red mug is located in only one of the compartments, the user needs to avoid opening the same compartment twice. This encourages careful planning and memory use to efficiently complete the task.

Example2:
  -Task:
  Task: After putting the butter and mug on the rack, pour in the milk, and check the wooden cabinet top region for any previously handled objects, placing them on the table if found.
  -Output Format:
  MEMORY RELATED GOAL:Check the wooden cabinet top region for any previously handled objects, placing them on the table if found.
  -Reason for Output(DO NOT output this part):
  To achieve the goal, the user must remember previously handled objects.

Please answer and ONLY ANSWER MEMORY RELATED GOAL!DO NOT OUTPUT REASON!

Here is the task:
{{task}}

Figure 12: **Memory Related Goal Generation Prompt.**

## Hierarchical Framework: Updating Plan & Memory

Analyze this image and determine if the following Memory related goal has been completed:

Current step: {step_text}

Previously completed steps: {completed_steps_text}

Memory related goal: {memory_related_goal}

Based on the image, has the Memory related goal been completed? Answer with 'Yes' or 'No' and explain your reasoning.

Note:

1. When you check cabinet top region for any previously handled objects,an object is considered previously handled if it also appears on the table.

2. When you check cabinet top region for any previously handled objects, consider the task complete if no objects are found inside the cabinet.

3. As long as you see any object in the opened cabinet, you can consider that you have found the item you need.

4. The butter looks like a red box.

5. The cream cheese looks like a blue box.

Figure 13: **Plan & Memory Updating Prompt.**

## Comparison of Plan Quality

***Task***: Place the plates neatly between the knife and fork, and pour all the snacks and drinks into the plates.

***GPT:***

...

Step: Place the popcorn box back to the far-right side of the table

Step: Pick up the wine bottle from the far-left side of the table

Step: Pour wine onto the plate positioned between the knife and fork

Step: Place the wine bottle back to the far-left side of the table ✅ *More complete action descriptions*

***Qwen:***

...

Step: Place the popcorn box back on the table.

Step: Pick up the wine bottle.

Step: Pour the wine into the plate.

***LLaVA :***

Step: Pick up the dining set from the wooden cabinet. ❌ *Unnecessary steps*

Step: Place the dining set on the table. ❌ *Unnecessary steps*

...

Step: Arrange the plates neatly between the knife and fork. ❌ *Incomplete decomposition*

Figure 14: **Comparison of Planning Quality.**

**Comparison of Plan Quality**

**Task:** Place the cookies into the basket, pour the tomato sauce and BBQ sauce into the frying pan, pour the wine into the red coffee mug, and place the white-yellow mug into the wooden tray.

*GPT:*
Step: Pick up the cookies from the table
Step: Place the cookies into the basket on the table
Step: Pick up the tomato sauce from the table
Step: Pour the tomato sauce into the frypan on the table
Step: Place the tomato sauce back on the table ✅ *Maintains completeness in long sequences*
...
Step: Pick up the white-yellow mug from the table
Step: Place the white-yellow mug into the wooden tray on the table

*Qwen:*
Step: Pick up the cookies from the table.
Step: Place the cookies into the basket.
Step: Pick up the tomato sauce bottle from the table.
Step: Pour the tomato sauce into the frying pan. *Missing some steps*
...
Step: Pick up the wine bottle from the table.
Step: Pour the wine into the red coffee mug.
Step: Pick up the white-yellow mug from the table.

*LLaVA:*
Step: Open the wooden tray region ❌ *Incorrect steps*
Step: Pick up the white-yellow mug from the wooden tray ❌ *Incorrect steps*
Step: Pour the tomato sauce from the bottle into the frying pan
...
Step: Pick up the basket from the table ❌ *Incorrect steps*
Step: Place the cookies in the basket ❌ *Incorrect steps*
Step: Pour the white-yellow mug into the wooden tray ❌ *Incorrect steps*

Figure 15: **Comparison of Planning Quality.**

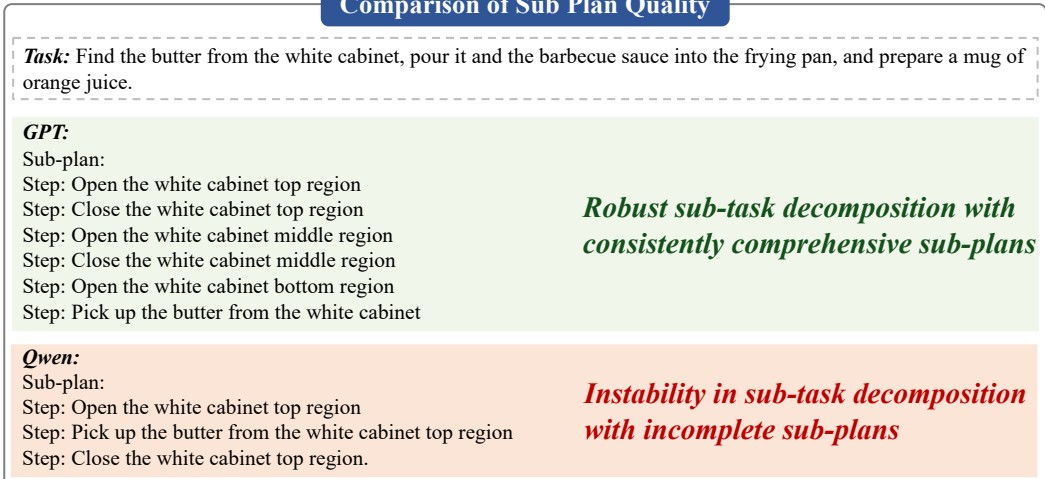

**Comparison of Sub Plan Quality**

**Task:** Find the butter from the white cabinet, pour it and the barbecue sauce into the frying pan, and prepare a mug of orange juice.

*GPT:*
Sub-plan:
Step: Open the white cabinet top region
Step: Close the white cabinet top region
Step: Open the white cabinet middle region
Step: Close the white cabinet middle region
Step: Open the white cabinet bottom region
Step: Pick up the butter from the white cabinet

*Robust sub-task decomposition with consistently comprehensive sub-plans*

*Qwen:*
Sub-plan:
Step: Open the white cabinet top region
Step: Pick up the butter from the white cabinet top region
Step: Close the white cabinet top region.

*Instability in sub-task decomposition with incomplete sub-plans*

Figure 16: **Comparison of Sub Plan Quality**

Figure 17: **Comparison of Task Completion Update Capabilities in Memory Exploration Task.**

## Comparison of Task Completion Update Capabilities in Memory Execution Task

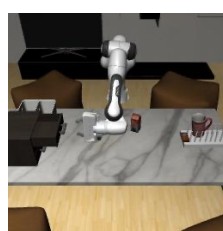

**Task:** After putting the butter and mug on the rack, pour in the milk, and check the wooden cabinet top region for any previously handled objects, placing them on the table if found.

*GPT:*
Step 1: ✅ *Correct inference in empty-cabinet scenarios*
Yes, the Memory related goal has been completed. The cabinet top region appears to be open, and there are no objects visible inside the cabinet. The task is considered complete if no objects are found inside the cabinet, which is the case here.

*Qwen:*
Step 1: ❌ *Difficulty recognizing empty-cabinet scenarios*
No
Reasoning:

...
- However, we do not know if the check for the top region of the cabinet was completed. There are no explicit signs, such as an empty cabinet or the placement of objects, to support the claim that the check was finished.
...
Therefore, based on the information provided in the image, the Memory related goal cannot be conclusively stated as completed.

Figure 18: **Comparison of Task Completion Update Capabilities in Memory Execution Task.**

