# OpenReview forum: "RoboCerebra: A Large-scale Benchmark for Long-horizon Robotic Manipulation Evaluation"
_NeurIPS.cc/2025/Datasets_and_Benchmarks_Track — NeurIPS 2025 Datasets and Benchmarks Track poster_

### Official Review · Reviewer_oLxF · 2025-06-15

**Rating:** 5
**Confidence:** 4

**Summary:**

The paper introduces RoboCerebra, a novel benchmark designed to evaluate System 2 reasoning capabilities—deliberative, high-level cognitive functions—in robotic manipulation tasks. The benchmark emphasizes long-horizon, compositional tasks that go beyond the short, reactive behaviors typically assessed in existing robotic datasets. RoboCerebra consists of:
1: A large-scale simulation dataset with 1,000 human-annotated trajectories across 100 task variants;
2: A hierarchical planning and execution framework combining a high-level vision-language model (VLM) planner and a low-level VLA controller;
3: A multi-dimensional evaluation protocol focusing on planning, reflection, and memory.
The authors benchmark several state-of-the-art VLMs (e.g., GPT-4o, Qwen2.5-VL) as System 2 modules and demonstrate improved performance under the proposed hierarchical framework.

**Additional Feedback:**

See the above Limitations.
Minor Suggestions:
1. A demo video showing a successful long-horizon execution would greatly enhance understanding;
2. Provide failure examples or qualitative analysis of where models fail;

**Dataset Code Accessibility:**

Yes

**Dataset Code Comments:**

The dataset and code are claimed to be openly available, with detailed instructions and modular pipelines. The paper documents a reproducible process for task generation, scene verification, and annotation.

**Ethical Considerations:**

Yes, there are ethics concerns that require attention by the authors

**Final Justification:**

I have carefully considered the authors’ response and reviewed the comments provided by the other reviewers, which are generally positive toward the manuscript. I will maintain my overall positive opinion of the paper.

**Limitations Weaknesses:**

1. Although the authors argue that high-level planning is less sensitive to the sim-to-real gap, the exclusive reliance on simulation limits the benchmark’s external validity. Real-world deployment often reveals unmodeled challenges.
2. Despite the introduction of multi-dimensional metrics, most quantitative comparisons still center around success rates. Other aspects like planning diversity, explanation quality, or failure recovery could be explored.
3. The hierarchical framework is relatively complex and hardware-intensive, requiring both fine-tuning and large-scale training (e.g., 200K steps on A100 GPUs). Accessibility for lower-resource labs may be limited.
4. While the tasks span 100 variants, they are still constrained to household simulations. Scenarios like industrial assembly, outdoor navigation, or social robotics would test broader reasoning capabilities.
5. There is limited discussion on potential biases in LLM-generated plans or instructions. Given LLMs’ known limitations (e.g., hallucination, cultural bias), it's important to assess how these affect benchmark integrity.

**Strengths Contributions:**

The paper addresses a critical gap in robotic learning benchmarks—long-horizon semantic reasoning. While many benchmarks test low-level motor control or reactive behavior, RoboCerebra is explicitly designed to probe deliberative planning, memory use, and reflection, making it a novel and much-needed contribution.
The dataset is impressive in scale and detail: over 10,000 step-level segments, rich annotations, multi-view verification, dynamic scene changes, and six task categories emphasizing memory, adaptability, and robustness. The 6× longer trajectories than prior benchmarks highlight its depth.
The comparison with existing datasets (Table 1) makes a strong case for RoboCerebra’s uniqueness, with clear evidence that no existing benchmark addresses all six features introduced here (e.g., LLM-generated tasks, human trajectories, dynamic environments, etc.).

---

> ### Author Rebuttal · Authors · 2025-07-31
>
> # Response to Reviewer oLxF
> We sincerely thank the reviewer for the comprehensive feedback. Below, we address each concern raised by the reviewer.
>
> > Although the authors argue that high-level planning is less sensitive to the sim-to-real gap, the exclusive reliance on simulation limits the benchmark’s external validity. Real-world deployment often reveals unmodeled challenges.
>
> Thank you for raising the crucial point of real-world validation. We completely agree and have a concrete roadmap to address it, designed specifically to test our core hypothesis.
>
> Our hypothesis posits that the sim-to-real gap is most pronounced at the level of the low-level controller (VLA), while the high-level planner (VLM) transfers more effectively due to its abstract nature. To provide direct evidence for this, we are currently building a real-world environment with a physical layout identical to our simulation.
>
> Our validation strategy, using a Franka Emika arm, is to first collect a real-world dataset to train the low-level VLA controller, thereby bridging the physical sim-to-real gap. With this real-world-adapted VLA in place, we will then deploy the high-level VLM planner, which was trained entirely in simulation, to execute long-horizon tasks. This setup allows us to cleanly isolate and validate the transferability of the VLM's abstract planning intelligence.
>
> We recognize that challenges exist, chief among them being the efficiency of data collection for the VLA. This will be a primary focus of our future work. We believe this plan offers a rigorous and transparent path to demonstrating the real-world viability of our approach and will substantially strengthen our work.
>
> > Despite the introduction of multi-dimensional metrics, most quantitative comparisons still center around success rates. Other aspects like planning diversity, explanation quality, or failure recovery could be explored.
>
> Thank you for your valuable suggestion. Following your advice, we have conducted a new analysis to evaluate the diversity and quality of the generated plans. For each task, we generated five distinct plans by sampling at a fixed temperature. We then used a GPT-based scoring system to assess both their diversity and quality, with each metric rated on a 1-to-10 scale. The results are presented in the table below. We acknowledge that any LLM-based evaluation may have inherent biases, and we plan to explore more comprehensive evaluation dimensions in our future work. To this end, a key direction is to construct a benchmark with more challenging failure scenarios to specifically test recovery capabilities. As shown in Figure 1 of our paper, our current work already includes initial failure recovery tasks (e.g., recovering a fallen wine bottle). Our future work will build upon this foundation to create a dedicated benchmark for a more thorough investigation of robustness.
>
> | VLM | Diversity (↑) | Quality (↑) |
> | :--- | :---: | :---: |
> | LLaVA-N-video | 4.00 | 4.69 |
> | LLaVA-N-video-blind | 4.14 | 4.89 |
> | Qwen2.5-VL | 5.31 | 5.74 |
> | Qwen2.5-LM | 5.77 | 6.11 |
> | GPT-4o | 7.00 | 6.37 |
> | GPT-4o-blind | 6.83 | 6.77 |
>
> > The hierarchical framework is relatively complex and hardware-intensive, requiring both fine-tuning and large-scale training (e.g., 200K steps on A100 GPUs). Accessibility for lower-resource labs may be limited.
>
> Thank you for your valuable feedback regarding computational resources and accessibility. We completely agree that this is a critical aspect for the broader adoption and reproducibility of research. In the paper, we reported a configuration using an A100-80G GPU with a large batch size of 64. This setup was chosen primarily to maximize training efficiency, which allowed for rapid iteration and validation of our methods. However, considering the varying hardware availability across different labs, we also conducted tests on a more accessible RTX 4090 GPU. Our experiments validate that the model can be trained successfully on this hardware by reducing the batch size to 8. Naturally, this entails a trade-off in training time; for the same number of training steps, the total training duration increases by approximately eightfold. We believe this viable option on more accessible hardware provides an effective alternative path for researchers with more limited computational resources.
>
> > While the tasks span 100 variants, they are still constrained to household simulations. Scenarios like industrial assembly, outdoor navigation, or social robotics would test broader reasoning capabilities.
>
> Our current work focuses on desktop robotic arm manipulation tasks specifically to validate the long-horizon planning and execution capabilities. We chose this setting because such scenarios naturally demand the kind of complex, step-by-step reasoning that is highly relevant to human daily activities. The scenarios you mentioned, such as industrial assembly and outdoor navigation, are indeed valuable directions for future extension. For instance, industrial settings demand extremely high precision and stability in repetitive execution, while outdoor environments introduce dynamic and unpredictable challenges. These unique challenges represent an important component of our future research plan, which we will actively explore.
>
> > There is limited discussion on potential biases in LLM-generated plans or instructions. Given LLMs’ known limitations (e.g., hallucination, cultural bias), it's important to assess how these affect benchmark integrity.
>
> Thank you for this question. It's important to distinguish between the effect of hallucinations on our dataset construction versus on the LLM's performance as a planner. **(1) On Dataset and Benchmark Construction:** To ensure the quality of our dataset and benchmark, we implemented a rigorous two-stage filtering process to address potential LLM hallucinations. First, we utilized the state-of-the-art GPT-o3-mini model with advanced prompts that included built-in self-checking and multiple refinement rounds, allowing us to automatically filter most logical inconsistencies. Subsequently, our team conducted a meticulous human verification process, where all sub-task decompositions and final trajectory annotations were manually inspected and corrected to remove any remaining errors. This robust, two-layer approach ensures that our final dataset is of high quality and integrity.
> **(2) On LLM Performance as a Planner:** Regarding the performance of LLMs as planners, our results confirm that hallucinations indeed affect their planning capabilities. Our review of several hallucination leaderboards shows that GPT-4o is a top-performing model in terms of factual accuracy and mitigating illusion. This finding correlates with the results in our paper; in Tables 3 and 4, GPT-4o demonstrates superior overall performance compared to models like Qwen-VL-Max and LLaVA-Next-Video. This strongly suggests that a model's propensity for hallucination negatively impacts the quality of its planning. To further illustrate this point, we provide several qualitative examples below that showcase the differences in planning performance across these models:
> - LLaVA-Next-Video: "Make sure to: Identify all relevant objects..."(unrelated meta-instructions).
> - LLaVA-Next-Video: "Place the cookies in the white bowl...Pour the cookies into the white bowl..." (repeated unnecessarily).
> - Qwen2.5-VL: "Open the top region of the short cabinet...Open the top region of the short cabinet..." (repeated unnecessarily).

---

> > ### Comment · Reviewer_oLxF · 2025-08-06
> >
> > Thank you for the author's response. I acknowledge that there is still a long way to go in developing a realistic and effective simulation environment, as well as certain limitations in its implementation.
> > I will maintain a positive attitude toward the authors' work.

---

### Official Review · Reviewer_CY8m · 2025-06-30

**Rating:** 5
**Confidence:** 4

**Summary:**

This paper introduces RoboCerebra, a large-scale benchmark designed to evaluate "System 2" capabilities in long-horizon robotic manipulation. By focusing on complex, multi-step tasks in dynamic environments, it aims to address limitations of prior reactive benchmarks and push the development of more deliberative robotic planners. The benchmark proposes a hierarchical framework combining a high-level VLM planner with a low-level VLA controller, along with an evaluation protocol targeting planning, reflection, and memory.

**Dataset Code Accessibility:**

Yes

**Ethical Considerations:**

No, there are no or only very minor ethics concerns

**Final Justification:**

RoboCerebra effectively addresses a key gap in current robotic manipulation benchmarks by targeting "System 2" reasoning—emphasizing long-horizon tasks, diverse subtask sequences, and dynamic environments, which are often overlooked in existing "System 1"-focused benchmarks.

1. The additional experiments and detailed analysis have sufficiently addressed my concerns.

2. I encourage the inclusion of more demonstration examples in future versions to further strengthen the benchmark.

Given the importance of this contribution and the authors’ strong response, I maintain my positive score: 5 (Accept).

**Limitations Weaknesses:**

1. While the paper evaluates a selection of state-of-the-art VLMs, the benchmark's model evaluation could be more comprehensive by including other notable embodied-specific models, such as RoboBrain, RoboMamba, Cosmos-reason1, and SpatialVLM.
2. This is a excellent and thoughtful work. RoboCerebra presents a compelling benchmark for evaluating "System 2" capabilities in long-horizon robotic manipulation, primarily by having the high-level VLM output text step-instructions that are then executed by a pre-trained VLA.  Could the authors elaborate on the specific rationale for choosing this text-based intermediate representation between the high-level planner and the low-level executor, as opposed to a visual grounding approach (e.g., the upper layer outputting bounding boxes/points for the lower layer to perform motion planning or use an oracle engine)?

**Strengths Contributions:**

1. RoboCerebra is specifically designed to address this gap that most existing robotic manipulation benchmarks primarily focus on "System 1" by focusing on long-horizon tasks, diverse subtask sequences, and dynamic environments.
2. The benchmark proposes a sophisticated top-down data generation pipeline.
3. Beyond simple task success, RoboCerebra introduces a multi-dimensional evaluation protocol. It includes four complementary metrics, providing a more comprehensive understanding of System 2 behavior.

---

> ### Author Rebuttal · Authors · 2025-07-31
>
> # Response to Reviewer CY8m
> We sincerely thank the reviewer for your valuable time.
>
> > While the paper evaluates a selection of state-of-the-art VLMs, the benchmark's model evaluation could be more comprehensive by including other notable embodied-specific models, such as RoboBrain, RoboMamba, Cosmos-reason1, and SpatialVLM.
>
> Thank you for your valuable suggestion. We performed additional experiments with the models you indicated. (*All VLA models use OpenVLA.*) The results are presented below:
>
> | Method | Para.| *Avg* | *Ran.* | *Obs.* | *Exp.* | *Exe.* | *Mix* | *Ideal* |
> | :--- | :---: | :---: |:---: | :---: | :---: | :---: | :---: | :---: |
> | LLaVA-N-Blind | 7B | 8.00 | 13.97 | 12.33 | 3.54 | 3.54 | 0.37 | 14.25 |
> | Cosmos-Reason1 | 7B | 8.41 | 7.63 | 10.45 | 5.55 | 7.31 | 8.73 | 10.79 |
> | VeBrain | 8B | 9.41 | 12.89 | 12.35 | 7.06 | 3.65 | 4.21 | 16.32 |
> | Qwen2.5-VL | 7B | 11.19 | 14.25 | 14.25 | 2.63 | 12.61 | 6.67 | 16.71 |
> | LLaVA-N-Video | 7B | 11.37 | 16.71 | 16.16 | 1.07 | 10.87 | 3.70 | 19.73 |
> | RoboBrain-2.0 | 7B | 11.40 | 12.11 | 12.06 | 9.92 | 10.96 | 7.27 | 16.05 |
> | GPT-4o | - | 16.04 | 18.63 | 19.45 | 8.04 | 16.69 | 11.48 | 21.92 |
>
> The results indicate that VLMs specifically designed for embodied tasks, such as RoboBrain-2.0, Cosmos-Reason1 and VeBrain, tend to outperform general-purpose VLMs on Embodied-QA benchmarks. This specialized proficiency translates into a notable advantage on the complex tasks we constructed, particularly in complex scenarios like Memory Exploration (Exp.) and Mix. However, this capability has not yet fully generalized to the domain of Embodied Planning, highlighting a crucial direction for future research in this field. An example is shown as follows:
> - Cosmos-Reason1: "Position the plate centrally between the knife and fork in the dining set group" (no discrete atomic actions and missing prior pick action).
> - VeBrain: "Move the plate backward slightly...Move the fork forward slightly...Move the knife backward slightly..." (repetitive adjustments without clear purpose)
> - RoboBrain-2.0: "Place the cookies into the basket," directly echoing the task instruction without decomposition (missing prior pick action and no discrete atomic actions).
> - GPT-4o: "Pick up the cookies from the table" followed by "Place the cookies in the basket..." (clear atomic breakdown).
>
> > This is a excellent and thoughtful work. RoboCerebra presents a compelling benchmark for evaluating "System 2" capabilities in long-horizon robotic manipulation, primarily by having the high-level VLM output text step-instructions that are then executed by a pre-trained VLA. Could the authors elaborate on the specific rationale for choosing this text-based intermediate representation between the high-level planner and the low-level executor, as opposed to a visual grounding approach (e.g., the upper layer outputting bounding boxes/points for the lower layer to perform motion planning or use an oracle engine)?
>
> We thank the reviewer for their positive assessment and for raising this insightful question. We chose **text-based intermediate representations** over visual primitives (e.g., bounding boxes or points) for two main reasons: **compatibility and feasibility**, and **support for hierarchical decomposition**.
>
> **(1) Compatibility and feasibility:** Our benchmark focuses on using the VLM as the **System 2** module, which is naturally better at producing text-based planning signals. At the same time, most existing VLAs (System 1) already accept language instructions as inputs alongside visual observations. Using text ensures broad compatibility without additional model modifications. In contrast, visual primitives present unresolved challenges: training a VLM to reliably produce accurate bounding boxes or points is non-trivial, and there is no standardized way for VLAs to robustly interpret these markers. Text-based instructions leverage a mature and widely adopted technology stack, making them a more reliable choice for the benchmark.
>
> **(2) Hierarchical task decomposition:** Our benchmark is designed to evaluate long-horizon reasoning using a hierarchical framework. Text-based single-step commands naturally serve as low-level building blocks that can be composed into complex tasks. This mirrors the **System 2** cognitive process, where complex goals are broken down into smaller, actionable steps.
>
> For example, consider the task *“pick an apple from a pile.”*
>
> * **Visual primitives:** The planner would need to choose a specific apple using a bounding box. In cluttered scenes, this may be ambiguous or overly restrictive (e.g., selecting an occluded apple).
> * **Text instructions:** A command such as *“pick up an apple”* leaves the choice of which apple to the VLA, which can leverage its perception and grasping abilities to select the most feasible option, resulting in more robust execution.
>
> We believe that combining visual and text-based signals may ultimately be the most effective way to enable rich interaction between **System 2** (VLM) and **System 1** (VLA). We plan to further explore this promising direction in our future work.

---

> > ### Comment · Reviewer_CY8m · 2025-08-05
> >
> > I would like to thank the authors for their responses. The additional experiments and detailed analysis have sufficiently addressed my concerns.
> >
> > Regarding the second point, I’d like to further discuss a few details (**no need for additional experiments due to time constraints**—I'm looking forward to seeing **more concrete demos** in future versions). Specifically, does the planner generate all step-wise instructions in a single pass? If so, in an open-loop setup, how is success or failure detected, and how is error recovery handled? Alternatively, if the system is designed to operate in a closed-loop manner, does the model adaptively revise plans based on historical observations?
> >
> > It would be interesting to see future work include more detailed metrics and analysis on aspects such as hallucination, success/error detection, recovery, and replanning. These could provide deeper insights into the robustness and reliability of the system.

---

> > > ### Comment · Reviewer_CY8m · 2025-08-05
> > >
> > > (One last question and no need for additional experiments—hope you don’t mind!) This is a very meaningful and well-executed contribution. I’m curious about the decision to use a VLA as the executor. Given that alternative language-conditioned policies or oracle engines can be more efficient and require less fine-tuning, it would be great to hear more about the reasoning behind this choice.

---

> > > > ### Author Response · Authors · 2025-08-05
> > > >
> > > > > I’m curious about the decision to use a VLA as the executor. Given that alternative language-conditioned policies or oracle engines can be more efficient and require less fine-tuning, it would be great to hear more about the reasoning behind this choice.
> > > >
> > > > While oracle engines work well for simple tasks like pick-and-place, they struggle with the complexity of real-world daily manipulation. Tasks that require fine-grained control, such as opening or closing doors, folding clothes, or handling articulated objects, cannot be accurately modeled using only discrete coordinate points. These tasks involve subtle motion trajectories, contact forces, and continuous changes in state, which go beyond what low-dimensional spatial coordinates can represent. Because oracle engines rely on simplified state inputs, they are inherently limited in capturing this level of detail.
> > > >
> > > > Long-horizon tasks, such as "open the cabinet, pick up the mug, and close the cabinet," further expose the limitations of coordinate-based control. In these settings, coordinate signals may appear similar across different steps, making it difficult to determine the correct timing and sequence of actions. Aligning spatial coordinates with temporal order requires precise coordination, which is error-prone and hard to generalize.
> > > >
> > > > In contrast, a VLA executor uses language understanding to capture task semantics and generate context-aware action sequences. This makes it better suited for handling the complexity, ambiguity, and variability of real-world robotic tasks.
> > > >
> > > > If you have any further questions or comments, please feel free to reach out to us.

---

> ### Author Response · Authors · 2025-08-05
>
> Thank you for your valuable response and insights.
>
> > Does the planner generate all step-wise instructions in a single pass?
>
> Yes, to ensure fair comparison, all planners currently generate instructions in a single pass. Admittedly, we believe planners could achieve better performance if multiple inferences are conducted and selections are made based on specific strategies.
>
> > If so, in an open-loop setup, how is success or failure detected, and how is error recovery handled?
>
> The open-loop setting is less robust than the closed-loop setup. During step-by-step execution, the system continues to the next step even if a previous step fails, without attempting to recover or correct the error.
>
>
> > Alternatively, if the system is designed to operate in a closed-loop manner, does the model adaptively revise plans based on historical observations?
>
> Yes. In the current version, we design prompts that guide the VLM to determine whether the current step has been successfully completed, with the output being either "True" or "False". If the result is "False", the system attempts to re-execute the current step. To support this mechanism, we include a small number of failure-recovery samples in the training data to ensure the VLA model can handle such scenarios.
>
> For example, in the *Memory Exploration* task where the subtask is "take a mug from the cabinet", our HPE framework (shown in Fig. 4) leverages previous observations. If the mug has already been retrieved from the second shelf, the model removes redundant instructions such as "open the third shelf of the cabinet".
>
> > It would be interesting to see future work include more detailed metrics and analysis on aspects such as hallucination, success/error detection, recovery, and replanning. These could provide deeper insights into the robustness and reliability of the system.
>
> Thank you for the suggestions. We will explore these aspects in our future work.

---

### Official Review · Reviewer_3eai · 2025-07-01

**Rating:** 5
**Confidence:** 3

**Summary:**

This paper focuses on benchmarking VLM’s capability in terms of semantic reasoning and long-horizon planning for the manipulation task. They introduce RoboCerebra, a benchmark for evaluating high-level reasoning in long-horizon robotic manipulation. The dataset generation procedural are clearly introduced, and they proposed a novel evaluation protocol with recent VLMs covered in the experiments.

**Dataset Code Accessibility:**

Yes

**Dataset Code Comments:**

the dataset is accessible via the link provided for inspection.

**Ethical Considerations:**

No, there are no or only very minor ethics concerns

**Final Justification:**

All raised concerns or questions have been clearly responded. I do believe this work deserves to be shared with the community for its timeliness and good thoroughness. The rebuttal content if possible, should be integrated into the revised paper, particularly those efforts to ensure the dataset quality. This paper makes a good contribution to the conference.

**Limitations Weaknesses:**

Some minor points:
1. Tasks generation takes inputs of the set of sampled objects, and they are converted into structured representation, including affordances and spatial context, etc. Are those objects already sampled from some existing scenes? I am wondering where those info, e.g. spatial context, comes from.
2. L133, authors try to guide the LLM to generate the tasks/steps following some conditions, but how well LLM follow the guidance? I would imagine it is still challenging for LLMs to understand physical and logical feasibility.
3. The human demonstration and annotation takes 400-hr effort. How many humans are involved? Are the humans indeed ensuring higher quality? Some more details on this part would be great, and I think they deserve to stay in the main paper.
4. Why only use OpenVLA as the VLA? There are other open-source models, such as https://huggingface.co/blog/pi0, or https://github.com/PKU-HMI-Lab/Hybrid-VLA, some justification on this will be good.

**Strengths Contributions:**

1. The study is timely, in line with the community’s interests.
2. Compared to existing benchmarks, the training scale is much larger, and the trajectory length is also significantly longer.
3. The quality control design is well-thought and interesting. I particularly like the idea of using the scene initialisation and verification execution via simulation.
4. The evaluation protocol reflects the purpose of the benchmark.

---

> ### Author Rebuttal · Authors · 2025-07-31
>
> # Response to Reviewer 3eai
>
> We sincerely thank the reviewer for the comprehensive and encouraging feedback. Below, we address each concern raised by the reviewer.
>
> > Tasks generation takes inputs of the set of sampled objects, and they are converted into structured representation, including affordances and spatial context, etc. Are those objects already sampled from some existing scenes? I am wondering where those info, e.g. spatial context, comes from.
>
> Thank you for your insightful question. To ensure the physical and logical plausibility of these automatically generated scenes, we employ a dual mechanism: **(1) Pre-defined Affordance and Volumetric Constraints:** We predefine the affordances and volumes of all objects. This foundational setup serves to eliminate basic logical inconsistencies and prevent spatial overlaps, as it establishes clear boundaries and interaction rules that objects must adhere to from the outset. **(2) VLM-Powered Common Sense Validation:** We integrate a VLM into our automated generation pipeline. Specifically, we input multi-view images of the scene into GPT-4o alongside carefully crafted prompts, tasking it with visually judging whether the object arrangement aligns with common sense. The model is instructed to output only "yes" or "no" as its response.
>
> > L133, authors try to guide the LLM to generate the tasks/steps following some conditions, but how well LLM follow the guidance? I would imagine it is still challenging for LLMs to understand physical and logical feasibility.
>
> We agree with the reviewer on this important point and have taken measures to mitigate the potential impact of LLM hallucinations or instruction-following failures during data collection. We addressed this issue at two levels. First, at the data construction stage, we used the state-of-the-art GPT-o3-mini model and designed prompts with built-in self-checking mechanisms. These included multiple refinement rounds where the model was asked to re-evaluate and verify the logical consistency of its outputs. Second, we incorporated human validation at critical stages. Specifically, all sub-task decompositions and final trajectory annotations underwent manual verification to ensure logical soundness, with erroneous samples removed or corrected. This two-layer strategy was essential to maintaining high data quality.
>
> > The human demonstration and annotation takes 400-hr effort. How many humans are involved? Are the humans indeed ensuring higher quality? Some more details on this part would be great, and I think they deserve to stay in the main paper.
>
> Our annotation process involved 25 human annotators and was designed as a three-stage sequential workflow to ensure data quality and integrity. The first stage, **Trajectory Annotation**, was conducted by 10 expert annotators who collected approximately 300 hours of valid trajectory data through teleoperation. In the second stage, **Temporal Boundary Annotation**, a different group of 10 annotators spent 100 hours labeling the precise start and end points of each sub-task. Finally, in the **Manual Inspection** stage, a team of 5 annotators devoted 200 hours to thorough quality control. Any annotation that did not meet the quality standards was rejected and re-annotated. This structured pipeline was essential for producing accurate trajectory paths and precise temporal segmentations.
>
> > Why only use OpenVLA as the VLA? There are other open-source models, such as pi0, or Hybrid-VLA, some justification on this will be good.
>
> Our primary objective was to evaluate the long-horizon planning and memory capabilities (System 2) of the current SOTA VLM models. To ensure fairness, we fixed the VLA model (System 1) during experiments and focused on how different VLM “brains” generated planning outcomes and interacted with a consistent System 1 module.
>
> We appreciate the suggestion to evaluate additional System 1 models to further strengthen the study. Accordingly, we conducted supplementary experiments using π0-fast (following the openpi repository) as the System 1 module. As shown in the table below, the current VLM model achieves comparable performance under this setting, confirming that its System 2 component is not tied to a specific VLA architecture. We will include these results and expand the relevant discussion in the final revised version.
>
> | Method | *Avg* | *Ran.* | *Obs.* | *Exp.* | *Exe.* | *Mix* | *Ideal* |
> | :--- | :---: | :---: | :---: | :---: | :---: | :---: | :---: |
> | Qwen2.5-LM-OpenVLA | 11.87 | 18.90 | 12.88 | 7.02 | 10.87 | 2.55 | 18.90 |
> | Qwen2.5-LM-π0-fast | 11.47 | 16.32 | 11.94 | 9.92 | 8.27 | 4.73 | 17.63 |
> | LLaVA-N-Blind-OpenVLA | 8.00 | 13.97 | 12.33 | 3.54 | 3.54 | 0.37 | 14.25 |
> | LLaVA-N-Blind-π0-fast | 5.99 | 7.89 | 8.06 | 5.89 | 3.85 | 1.82 | 8.42 |
> | GPT-4o-Blind-OpenVLA | 15.10 | 20.00 | 17.03 | 7.02 | 16.09 | 10.48 | 20.00 |
> | GPT-4o-Blind-π0-fast | 13.63 | 14.47 | 15.82 | 12.27 | 12.69 | 11.27 | 15.26 |
> | Qwen2.5-VL-OpenVLA | 11.19 | 14.25 | 14.25 | 2.63 | 12.61 | 6.67 | 16.71 |
> | Qwen2.5-VL-π0-fast | 13.19 | 20.79 | 15.45 | 8.24 | 6.15 | 6.91 | 21.58 |
> | LLaVA-N-video-OpenVLA | 11.37 | 16.71 | 16.16 | 1.07 | 10.87 | 3.70 | 19.73 |
> | LLaVA-N-video-π0-fast | 8.79 | 12.11 | 12.12 | 7.73 | 4.81 | 2.55 | 13.42 |
> | GPT-4o+OpenVLA | 16.04 | 18.63 | 19.45 | 8.04 | 16.69 | 11.48 | 21.92 |
> | GPT-4o+π0-fast | 15.15 | 18.95 | 20.00 | 10.59 | 11.73 | 10.18 | 19.47 |
> | GT-plan-OpenVLA | 25.16 | 26.85 | 30.68 | 19.47 | 23.48 | 19.26 | 31.23 |
> | GT-plan-π0-fast | 23.04 | 23.68 | 26.36 | 18.15 | 16.92 | 26.55 | 26.58 |

---

> > ### Comment · Reviewer_3eai · 2025-08-07
> > **Thx for the response**
> >
> > I sincerely appreciate authors' responses to my raised questions. All clear now for me. I do believe this work deserves to be shared with the community for its timeliness and good thoroughness. The rebuttal content if possible, should be integrated into the revised paper, particularly those efforts to ensure the dataset quality. I thus maintain my original rating, and I would like to champion this paper for an Oral presentation.

---

> > > ### Author Response · Authors · 2025-08-07
> > >
> > > We sincerely thank the reviewer for the positive feedback and for recognizing the value and timeliness of our work. We are pleased to hear that our responses have addressed the raised concerns. As recommended, we will incorporate the relevant content from the rebuttal into the revised version, particularly the details regarding our efforts to ensure the quality of the dataset.

---

### Official Review · Reviewer_bJWC · 2025-07-03

**Rating:** 4
**Confidence:** 4

**Summary:**

- This paper introduces RoboCerebra, a benchmark for evaluating high-level reasoning in long-horizon robotic manipulation.

- RoboCerebra includes: (1)a simulation dataset with extended task horizons and diverse subtask sequences, (2) a hierarchical framework combining high-level VLM planner with a low-level vision-language-action (VLA) controller. and (3) an evaluation protocol targeting planning, reflection, and memory.

**Dataset Code Accessibility:**

Partly

**Dataset Code Comments:**

I don't seem to see open-source datasets, evaluation environments, and code.

**Ethical Considerations:**

No, there are no or only very minor ethics concerns

**Final Justification:**

I maintain my initial positive rating. I think it would be valuable to the community if the authors could open source the data, code, and baselines.

**Limitations Weaknesses:**

- A major drawback is the paper's lack of specific design and corresponding contributions at the methodological level. The authors primarily fine-tuned existing System1 and System2 models with their own data, consequently focusing on evaluating System2's capabilities. Although submitted to the dataset track, it would have been better if the authors had introduced specific methodological designs based on issues discovered during the evaluation, thereby differentiating it from classic hierarchical designs (e.g., PALM-E and a series of subsequent works).

- How is the memory bank used during the replanning process implemented? It seems that the LLM's inherent in-context learning capabilities already achieve this . Did the authors perform any ablation studies on this module?

- Research utilizing large-scale multimodal models to enhance robotic systems' cognitive, reasoning, or planning capabilities is already extensive, and this paper appears to have omitted significant discussion, for instance, regarding works such as those listed below . I suggest the authors discuss representative works in subsequent versions.

[1] AlphaBlock: Embodied Finetuning for Vision-Language Reasoning in Robot Manipulation

[2] Physically grounded vision-language models for robotic manipulation

[3] ReplanVLM: Replanning robotic tasks with visual language models

[4] Transferring foundation models for generalizable robotic manipulation

[5] Embodiedgpt: Vision-language pre-training via embodied chain of thought

[6] Look Before You Leap: Unveiling the Power of GPT-4V in Robotic Vision-Language Planning

[7] OK-Robot: What Really Matters in Integrating Open-Knowledge Models for Robotics

**Strengths Contributions:**

- This paper constructs numerous long-horizon tasks and corresponding trajectory data based on LIBERO, which is highly valuable for the community in both training and evaluating System1 and System2 capabilities.

- This paper elaborately evaluates System2 capabilities, which is also very important.

---

> ### Author Rebuttal · Authors · 2025-07-31
>
> # Response to Reviewer bJWC
>
> We sincerely thank the reviewer for the encouraging feedback. Below, we address each concern raised by the reviewer.
>
> > A major drawback is the paper's lack of specific design and corresponding contributions at the methodological level. The authors primarily fine-tuned existing System1 and System2 models with their own data, consequently focusing on evaluating System2's capabilities. Although submitted to the dataset track, it would have been better if the authors had introduced specific methodological designs based on issues discovered during the evaluation, thereby differentiating it from classic hierarchical designs (e.g., PALM-E and a series of subsequent works).
>
> We appreciate the reviewer’s feedback regarding the need for a clearer explanation of our methodological design. The primary focus of this work is to evaluate and analyze the performance of the **System 2** (VLM) component. To this end, we deliberately adopted a **hierarchical framework** that decouples **System 2** (VLM) and **System 1** (VLA) modules. This design enables efficient plug-and-play evaluation of different System 2 models while preserving the original System 1 architecture as much as possible.
>
> We acknowledge that a **unified model** integrating both components is a valid alternative. However, we believe this approach is less suitable for isolating and evaluating System 2 performance, as it requires fully re-training the combined model for each variation, which is computationally expensive and less efficient. To validate this point, we compared our hierarchical framework against two baselines:
>
> **(1) Naive System 1-only model:** When trained directly on the long-horizon tasks, this model performed poorly in complex scenes and long sequences, demonstrating the need for a dedicated high-level planning module.
> **(2) Unified model:** We implemented a unified design by bridging a System 2 and a System 1 model using a feature projector. Although more end-to-end, this approach suffered from unstable training and exhibited approximately a *4%* performance drop compared to our hierarchical framework.
>
> These results highlight that our hierarchical framework not only allows for efficient evaluation of different System 2 models but also achieves superior performance. We attribute this advantage to its clear separation of roles: System 2 generates coherent, high-level task plans using its advanced reasoning abilities, while System 1 excels at executing precise low-level control steps. This decoupling ensures both stability and strong task performance. Moving forward, we are exploring more end-to-end architectures (e.g., Mixture-of-Transformers and World Models) to further enhance the collaboration between System 2 and System 1 components.
>
> > How is the memory bank used during the replanning process implemented? It seems that the LLM's inherent in-context learning capabilities already achieve this. Did the authors perform any ablation studies on this module?
>
> Thanks for your comments. The key difference between our memory bank and the standard in-context learning of an LLM is how past information is managed. A conventional LLM simply processes a flat, concatenated history within its context window. In contrast, our memory bank is an external, structured module that keeps track of the task state as the robot interacts with the environment.
>
> Here is how it works: when the robot executes an action (e.g., *“open the first cabinet”*), the memory bank explicitly records the outcome (success or failure) and any important observations (e.g., *“no cup inside”*). During replanning, instead of feeding the entire raw history back to the LLM, our system queries the memory bank to build a structured prompt that includes: **(i) the overall goal, (ii) all executed actions and their outcomes, and (iii) the list of remaining actions that have not yet been tried.**
>
> This state-aware context offers two key benefits. First, it avoids redundant commands by marking failed paths, preventing the LLM from repeating actions such as reopening an empty cabinet. Second, once the goal is achieved and recorded, the system can naturally terminate the task, improving overall efficiency.
>
> In short, our memory bank acts as a dynamically updated world model grounded in real physical feedback. This allows the LLM to operate as a stateful decision-maker, rather than a stateless reasoner prone to repeating mistakes, and significantly improves its ability to solve complex, long-horizon tasks.
>
> Below is an ablation study on the memory bank. We found that it provides a significant performance boost in memory-intensive tasks, namely Memory Exploration (Exp.), Memory Execution (Exe.), and Mix, confirming its crucial role in overall capability.
> | Method | *Avg* | *Ran.* | *Obs.* | *Exp.* | *Exe.* | *Mix* | *Ideal* |
> | :--- | :---: | :---: | :---: | :---: | :---: | :---: | :---: |
> | w/o Memory Bank | 16.04 | 18.63 | 19.45 | 8.04 | 16.69 | 11.48 | 21.92 |
> | w/ Memory Bank | 16.55 (+0.51) | 18.63 | 19.18 (-0.27) | 9.06 (+1.02) | 17.83 (+1.14) | 13.21 (+1.73) | 21.10 (-0.82) |
>
> > Research utilizing large-scale multimodal models to enhance robotic systems' cognitive, reasoning, or planning capabilities is already extensive, and this paper appears to have omitted significant discussion, for instance, regarding works such as those listed below. I suggest the authors discuss representative works in subsequent versions.
>
> Thank you for this valuable suggestion. We will include the following discussion of the representative works you mentioned in the revised version manuscript:
>
> Recent works have made significant strides in vision-language planning for robotics. Many of these pioneering systems, including AlphaBlock and ReplanVLM, rely on modular pipelines that combine LLMs with separate perception or control modules, rather than integrated Vision-Language-Action models. However, a primary limitation across these diverse architectures is the absence of a standardized and challenging benchmark, making it difficult to rigorously compare their merits, especially on long-horizon tasks. RoboCerebra addresses this critical gap by introducing a comprehensive benchmark tailored for evaluating modern planners. Its key advantage is enabling fair and precise quantitative comparison across different paradigms, particularly for the emerging class of VLA-based agents, thereby accelerating the development of robust robotic systems.
>
>
> > I don't seem to see open-source datasets, evaluation environments, and code.
>
> Thank you for your comment. We would like to clarify that the dataset and evaluation code were included with our submission. We would be grateful if you could take another look at the submitted version.

---

> > ### Comment · Reviewer_bJWC · 2025-08-03
> >
> > I would like to thank the authors for their responses. I noticed that the link to the evaluation script provided by the authors seems to be invalid. In addition, I would like to know whether the authors uploaded part of the data or all of the data?

---

> > > ### Author Response · Authors · 2025-08-03
> > >
> > > Thank you for your reply. We have carefully inspected the data and evaluation files, and the detailed information is as follows:
> > > (1) The data we submitted is complete. Each pair contains trajectories, environment initialization files, language instructions, temporal annotations, demonstration videos, images of the initial state, distractor objects, and task summaries. The training set consists of a total of 1006 pairs.
> > > (2) Our evaluation scripts were submitted in the form of supplementary materials.

---

> > > > ### Comment · Reviewer_bJWC · 2025-08-04
> > > >
> > > > Thank you to the authors for their further responses! I maintain my positive opinion.

---

> > > > > ### Author Response · Authors · 2025-08-04
> > > > >
> > > > > Thank you for your positive opinion. We will release the data, code, and baselines.

---

### Note · Authors · 2025-08-13

We thank the AC and all reviewers for their time and thoughtful feedback. The comments and in-depth discussions helped us improve the paper and inspired promising future directions.

Following the rebuttal and author-reviewer discussions, reviewers reached a clear and positive consensus on the merits of our work. They consistently recognized these key strengths:

- **Novelty and Critical Contribution:** Our paper introduces RoboCerebra, a benchmark targeting long-horizon reasoning and System 2 capabilities in robotics, which reviewers highlighted as a timely and needed addition to the community. [3eai, CY8m, oLxF]

- **Impressive Scale and Data Quality:** The dataset was praised for its scale, long, diverse trajectories, and rigorous annotation process. Reviewers commended the sophistication and care in our data generation and validation pipeline. [bJWC, 3eai, CY8m, oLxF]

- **Comprehensive Evaluation Protocol:** Reviewers appreciated our multi-dimensional evaluation that moves beyond task success, for deeper assessment of an agent’s planning and reasoning behavior. [bJWC, 3eai, CY8m]

We also addressed the remaining concerns, summarized as follows:

- **Methodological Design:** We clarified the rationale for our hierarchical framework [bJWC] and the use of text-based interfaces [CY8m], which support modularity and enable systematic evaluation of System 2 capabilities. Ablations show superior stability and performance compared to end-to-end baselines.

- **Expanded Experiments:** In response to calls for broader evaluations [CY8m, 3eai, oLxF], we added results with additional VLMs and VLAs (e.g., RoboBrain-2.0, Cosmos-Reason1, π0-VLA), new ablations demonstrating the importance of the memory bank, and analysis of planning quality and diversity.

- **Data Quality and Bias Mitigation:** To address concerns about LLM hallucinations and annotation robustness [3eai, oLxF], we described a validation pipeline combining VLM-based checks and 600+ hours of human verification. We also confirmed that data and code were included in the submission. [bJWC]

- **Scope and Real-World Applicability:** We acknowledged the limitations raised [oLxF] and outlined our roadmap for real-world validation on physical robots. We also clarified that our models can be trained on accessible hardware with manageable trade-offs.

We will incorporate the reviewers' suggestions into future revisions and follow-up research. Thank you again for your thoughtful engagement with our work.

---

### Decision · Program_Chairs · 2025-09-18

**Decision:**

Accept (poster)

**Comment:**

(a) The paper proposes a new benchmark for long-horizon reasoning in robotics.

(b) The benchmark tackles a very important problem. The paper is well written. The benchmark is very complete in its components (dataset, planner + controller, evaluation protocol) and has an extensive number of tasks. The benchmark is well designed with a clear focus.

(c) A few methodological details were unclear, the embedding in the state of the art needed to be improved, limited level 1 model evaluation, some design choices were unclear.

(d) After the rebuttal and discussion all major issues have been resolved and I fully agree with the reviewers that the benchmark is a very valuable contribution to the community and that the paper has significant added value.

(e) The reviewers highly appreciated the detailed replies and many additional results. There had been some misunderstandings that got resolved. Now all reviewers are convinced that the paper should be accepted.

===== FINAL UPDATE FROM DB Track PCs ====

The final decision for this paper has been taken by the program chairs after consultation with the SACs. All Senior Area Chairs have ranked papers according to the feedback from the AC during the review process. We decided to leave the original meta-review to reflect the opinion of the AC in light of the initial discussions with reviewers and SAC.